# Climate change, not human population growth, correlates with Late Quaternary megafauna declines in North America

Mathew Stewart [1,4✉], W. Christopher Carleton [1,4✉] & Huw S. Groucutt [1,2,3✉]

The disappearance of many North American megafauna at the end of the Pleistocene is a contentious topic. While the proposed causes for megafaunal extinction are varied, most researchers fall into three broad camps emphasizing human overhunting, climate change, or some combination of the two. Understanding the cause of megafaunal extinctions requires the analysis of through-time relationships between climate change and megafauna and human population dynamics. To do so, many researchers have used summed probability density functions (SPDFs) as a proxy for through-time fluctuations in human and megafauna population sizes. SPDFs, however, conflate process variation with the chronological uncertainty inherent in radiocarbon dates. Recently, a new Bayesian regression technique was developed that overcomes this problem—Radiocarbon-dated Event-Count (REC) Modelling. Here we employ REC models to test whether declines in North American megafauna species could be best explained by climate changes, increases in human population densities, or both, using the largest available database of megafauna and human radiocarbon dates. Our results suggest that there is currently no evidence for a persistent through-time relationship between human and megafauna population levels in North America. There is, however, evidence that decreases in global temperature correlated with megafauna population declines.

[1] Extreme Events Research Group, Max Planck Institutes for Chemical Ecology, the Science of Human History, and Biogeochemistry, Jena, Germany. [2] Department of Archaeology, Max Planck Institute for the Science of Human History, Jena, Germany. [3] Institute of Prehistoric Archaeology, University of Cologne, Cologne, Germany. [4]These authors contributed equally: Mathew Stewart, W. Christopher Carleton. ✉email: mstewart@ice.mpg.de; wcarleton@ice.mpg.de; hgroucutt@ice.mpg.de

Since its conception in the 1960's, Paul Martin's overkill hypothesis as an explanation for the extinction of most of North America's Late Quaternary megafauna (animals with an average adult body mass of ≥44 kg) has spurred a considerable amount of research and debate[1,2]. Near the end of the Pleistocene (~11,700 years before present [BP]) at least 37 genera of megafauna (~80%) had disappeared from North America, and by as early as the late eighteenth century[3,4] researchers were considering a human hand in the extinction of mammals in the continent. Martin later formalised this in his "overkill hypothesis", claiming that these extinctions were the direct result of overhunting of naïve prey by newly immigrated and rapidly expanding human populations at the close of the Pleistocene[5–7]. These extinctions may have been drawn-out over thousands of years, or, as the 'blitzkrieg' variant of overkill claims, occurred within centuries or less of human arrival[8,9]. Another variant, the 'sitzkrieg' model, suggests that alongside hunting, anthropogenically driven increases in fire, habitat fragmentation, and disease contributed significantly to the demise of North American megafauna[10]. Despite the variations, these overkill hypotheses all point to a correlation between increased hunting activity and megafauna extinctions and they are by far the most prominently discussed anthropogenic explanations for the Late Quaternary megafauna extinctions.

In contrast, other scholars consider the climatic and environmental changes associated with the end of the Pleistocene epoch to be the main driver of the megafauna extinctions rather than overhunting[2,11]. Arguments against overkill centre around (i) the scarcity of megafauna kill sites, which implies that humans were not hunting megafauna in sufficient numbers to drive them to extinction, and (ii) the fact that some megafauna last appearance datums (LADs)—i.e., the most recently dated fossil evidence for a given species—pre-date or significantly post-date human arrival to the Americas. At the same time, several lines of evidence point directly to the impact of past climate change on megafauna populations and ecology. Some ancient DNA studies, for instance, have shown that significant losses of genetic diversity for some taxa (e.g., bison) occurred prior to human arrival[12–14]. A number of bird and reptile taxa also went extinct[15], as did a species of spruce tree[16], while nine megafauna species survived and a new species of bison emerged[2]. Bison, bighorn sheep, elk, equids, and other taxa underwent significant reductions in body size[17–20], and there were extensive shifts in animal and plant ranges[21]. For some scholars, these details demonstrate that the North American megafauna extinction event was part of a drawn-out restructuring of the animal and plant communities driven by late Pleistocene climatic and environmental changes with humans playing at most a marginal role[2].

Radiocarbon dates indicative of extinction timing have been a key source of data for testing these hypotheses. First appearance datums (FADs)—the earliest dated fossil evidence for a given species—and LADs are often used in simple tests of the overkill hypothesis. If the LAD of a particular taxon pre-dates the FAD of humans, the logic goes, then the latter cannot be implicated in the extinction of the former. Conversely, if the LAD of a particular taxon postdates the FAD of humans, then it is possible that the latter played a decisive part in the extinction of the former.

There are, however, problems with FAD- and LAD-based studies. For instance, the LAD of *Smilodon fatalis* is, with near certainty, not derived from remains of the last living saber-tooth cat—a phenomenon known as the Signor-Lipps effect[22]. Even for extensively dated taxa, such as mammoth (*Mammuthus primigenius*), sedimentary ancient DNA studies have suggested that some taxa survived far beyond their LAD's based on dated fossil remains[23]. Consequently, LADs cannot provide a definite answer to even simple questions about the temporal coincidence of human arrival or climate change with megafauna extinctions.

Likewise, LAD-based studies cannot help us to understand the through-time dynamics of the extinction process. To robustly explore demographic change, including extinction, we require long-term population level time-series for both humans and megafauna, particularly if we aim to understand the relative roles of human activity and climate change in megafauna extinctions. Importantly, this includes assessing megafauna populations prior to human arrival as it is possible that some megafauna were already heading towards extinction by the time that humans arrived, with humans simply providing the final blow, or coup de grâce.

To overcome the limitations of appearances datums and investigate through-time population dynamics, some scholars have turned to a popular proxy for through-time past population levels: the summed probability density function (SPDF). SPDFs are summaries of the number of radiocarbon-dated events that occurred in each interval over some span of time (every decade between a given start and end date, for example). In the context of palaeodemography, they are often interpreted as if they represent through-time changes in population levels [e.g.,24,25]. With respect to studies of human population dynamics, the individual dates are derived from radiocarbon samples associated with evidence of human activity (e.g., hearth features, dated occupation layers, and so on) or directly from human skeletal remains. The number of such events in a given area dated to a given time is thought to correlate with the number of humans present in that area over that time[26]—more hearths probably means more people and more human skeletal remains almost certainly does. The same reasoning applies to radiocarbon dates associated with megafauna—more skeletal remains and other evidences like dung or hide probably means more megafauna present on the landscape[24].

SPDFs have been used a number of times in the study of megafauna extinctions. Boulanger and Lyman[25], for example, used them to argue that megafauna populations in the American Northeast were already in decline by the time humans arrived and identified earlier population level fluctuations which, they suggest, might be tied to increases in temperature, lake level fluctuations, and vegetation change. Mann et al.[27] used SPDFs to contend that Alaskan megafauna populations peaked during the initial phases of warm interstadials followed by population declines as peatlands spread. Similarly, MacDonald et al.[28] found declines in Beringian mammoth population levels which they correlate to the development of peatlands and reduction of grasslands. More recently, Broughton and Weitzel[24] took a taxon-specific approach in which they constructed SPDFs for humans and six well-dated North American megafauna taxa and compared those proxies to human SPDFs using linear regression. And in South America, SPDFs have been used to argue that megafauna populations were increasing throughout the Bølling-Allerød (B-A) until suddenly plummeting at the start of the Younger Dryas (YD)[29].

In most of these studies, the authors have acknowledged several well-known problems with the use of SPDFs. Important sources of bias include radiocarbon sample quality, the true chronological relationship between a given sample and its depositional context, spatio-temporal sampling adequacy, taphonomic processes (i.e., the degradation and loss of samples over time), and radiocarbondate calibration artefacts. These sources of bias can produce misleading results in analyses of SPDFs. Several papers have discussed these potential problems in detail and possible solutions have been offered[30,31].

In addition to these problems, SPDFs also conflate process variation with chronological uncertainty in a way that undermines their potential for analysing extinction dynamics[32]. SPDFs are simply a sum of radiocarbon-date densities. So, given two densities for example, any point on the SPDF curve is a combination of the number of events in question (two, in this example) and the

probability that each event dates to the relevant time[33–35]. The sum is treating information about chronological uncertainty—i.e., up-down fluctuations in the level of individual date densities—as if it directly reflects the number of events at a given time. Therefore, while SPDFs may be helpful tools for summarising chronological information or discerning certain patterns in large radiocarbon-date databases e.g.,[36], they are not an unambiguous indication of event-count or, by extension, a suitable proxy for population levels in a point-wise way[31,32,35].

This conflation has significant analytical consequences. When an SPDF is used as the response variable in a statistical regression intended to explain variation in event-count, the model is mis-specified by definition[32]. Rather than explaining variation in the number of events as a function of one or more covariates at a given time, the model is instead explaining variation in some inseparable combination of event-count and chronological uncertainty. The model would also be incapable of properly accounting for chronological uncertainty in its parameter esti-mates separately from sampling variability or real underlying process stochasticity. Attempting, then, to explain population fluctuations by comparing this proxy to some covariate (e.g., temperature, another SPDF, etc.) may lead to spurious correla-tions and faulty inferences. Indeed, recent simulation research has demonstrated this[31,32]. Consequently, previous research involving SPDFs may be giving a misleading impression of the available evidence regarding North American megafauna demographic responses to humans and climate change.

With this in mind, we use here a recently developed alternative—Radiocarbon-dated event count (REC) modelling[34]—to evaluate the North American megafauna overkill and climate change hypotheses. This new approach is a Bayesian regression technique that accounts for chronological uncertainty in time series of radio-carbon-dated event counts. It involves sampling alternate probable count sequences that are consistent with the uncertainties in the individual radiocarbon-date densities in a given database. A sample of alternate sequences—a Radiocarbon-dated Event Count Ensemble (RECE)—is first produced. Each sequence in the sample (RECE member) is then used as the response variable in a suitable regression model. The parameters estimated for these individual models are considered to be samples from a set of super-population parameter distributions that reflect the variability among the individual regression estimates. These individual model estimates vary because the alternate count sequences are all slightly different, reflecting the chronological uncertainty in the corresponding radiocarbon dates—a sequence of fossil counts, for example, might be {1,2,3} or {2,1,3} when the relevant fossil date uncertainties overlap. Thus, the super-population parameters of the model reflect chronological uncertainty as well. In effect, the REC model con-siders alternate histories, given the uncertainty in radiocarbon dates, and it uses those alternatives to estimate a set of super-population parameters (e.g., regression coefficients) that are con-sistent with the set of alternate histories (see the Methods section for further details).

We use this approach to analyse the most comprehensive published database of North American megafauna sample dates available (Fig. 1; data from Broughton and Weitzel[24]). In a series of analyses, we tested whether human population levels, climate change (using the NGRIP $\delta^{18}$O as a proxy for regional climate change), or both appeared to correspond quantitatively to through-time changes in megafauna population levels. We rea-son that if one or both could be implicated in the megafauna extinctions, the corresponding REC model regression coefficient (s) should differ significantly from zero—i.e., the posterior density estimates for the regression coefficients should exclude zero at the 95% confidence level or greater. Put differently, if human overkill drove megafauna extinctions, we expect there to

be a negative and statistically significant (non-zero) correlation between the human and megafauna population density proxies. Likewise, if rising temperatures drove megafauna extinctions, we expect a negative and statistically significant correlation between our megafauna population density and climate proxy, or, alter-natively, if decreasing temperatures caused megafauna extinc-tions, a positive correlation between these two proxies. Following a growing consensus on studying megafauna extinctions e.g.,[21,37,38], we created both models in which megafauna were treated collectively, and models in which megafauna were broken down by taxa and region (following Broughton and Weitzel[24]). We also accounted for potential taphonomic bias in the pub-lished fossil record by including an established proxy for taphonomic sample loss as a covariate in all models[39]. Our results suggest that there is currently no evidence for a persistent through-time relationship between human and megafauna population levels in North America. There is, however, evidence that decreases in global temperature correlated with megafauna population declines.

## Results

**Human, megafauna, and climate correlation analyses**. The posterior distributions from the regression models are shown in Figs. 2–4. The results of our analyses were all the same regarding the impact of human population size on megafauna population size. While controlling for taphonomy, the human population size proxy ($\beta_{Humans}$) was not significantly different from zero (denoted in each plot by a vertical grey line) in any analysis. This finding was consistent whether we aggregated all megafauna into a group, focused on individual taxa separately (Fig. 2), or broke the megafauna data down by taxa and region (Supplementary Fig. 1). Based on the available data, human population size, therefore, appears to have had no significant impact on mega-fauna population size from 15.0 to 11.7 ka.

Interestingly, the taphonomic proxy ($\beta_{TA}$) appears to have had no statistically significant effect in each of the models. This finding was again consistent whether the megafauna taxa were examined together, or separated by taxa and region (Fig. 2). Thus, over the ~4000-year period investigated, taphonomic processes do not appear to have created an obvious temporal trend in megafauna sample frequency.

In contrast, all of the models indicate that the climate change proxy ($\beta_{Climate}$)—i.e., the NGRIP $\delta^{18}$O record—and megafauna population size were correlated. In models involving only climate change, the regression coefficient for that proxy had a posterior mean estimate of at least 0.05 with most models having a posterior mean for that parameter of ~0.1 or higher (Fig. 3). The results were similar for models involving both the human population size proxy and the climate change proxy (Fig. 4). They were also the same in the regional analyses (Supplementary Figs. 2 and 3). Across all analyses performed, then, the only proxy that appeared to be correlated with the megafauna population size proxies was the NGRIP $\delta^{18}$O record. The effect size was fairly consistent—around 0.1 for most models—and positive.

**Extended climate analysis**. In light of these findings, we extended our climate analysis. Many recently published high-profile studies on Late Quaternary North American megafauna extinctions have relied on the ~50-year resolved NGRIP $\delta^{18}$O record e.g., refs. [24,28,40,41]. However, this record has two characteristics that limit its utility in quantitative analyses. Firstly, it is a heavily smoothed interpolation of the raw NGRIP $\delta^{18}$O time series; and, secondly, it comes with no indication of chronological uncer-tainty[42]. These characteristics imply that the patterns in the smoothed NGRIP record are likely biased. With that in mind, we

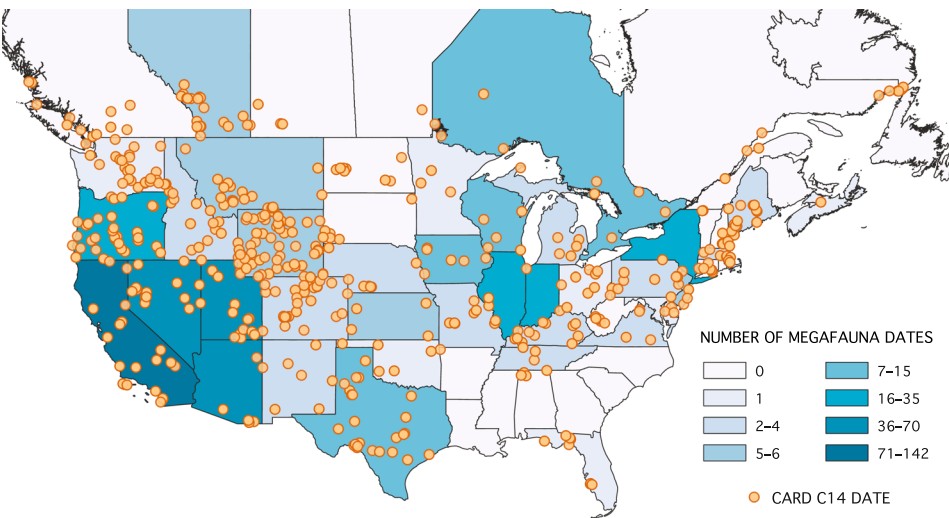

**Fig. 1 Map of the study area.** Contiguous United States and southern Canada with archaeological site locations (orange circles) and frequency of dated megafauna remains by state. Map recreated using data from Broughton and Weitzel[24] under the CC BY license 4.0.

performed an extended analysis using the raw annually-resolved NGRIP δ[18]O record[43] and accounted for chronological and measurement uncertainty in that record (Fig. 5 and see Supplementary Note 6). Additionally, we extended the study period to 20.0–10.0 ka, the beginning of which marks the start of the Northern Hemisphere deglaciation and the end of the LGM (following Clark et al.[44]). The reasoning being that if climate really was driving megafauna population dynamics, there should be a long history of that process observable in the available data.

The findings supported our primary results. The posterior mean of the top-level regression coefficient for the climate proxy was positive in all cases and close to the previous values (Fig. 6). Therefore, whether analysing all megafauna together, or separating by taxa and region, there is a consistent positive relationship with the NGRIP proxy record. That the findings between the smoothed and annually resolved NGRIP record are consistent suggests that the relationship between the megafauna and climate records is robust. Together, our primary and extended analyses suggest that humans, or more precisely that estimated changes in human population levels, had little bearing on North American megafauna population levels, but that decreases in global temperature had an overall negative impact on megafauna population levels.

## Discussion

Our results are at odds with simple overkill models that imply that multiple North American megafauna were directly driven to extinction by unsustainable hunting of rapidly expanding human populations[6]. Likewise, while recent studies have often emphasised that both overkill and climate change played a role in the extinction of different species of megafauna[24], our analysis failed to replicate this finding and instead found a consistent correlation only between climate change and North American megafauna population levels. This was the case regardless of whether we analysed all megafauna together, or separated megafauna by taxa and region. It remained the case in our extended analysis involving the annually-resolved NGRIP climate proxy for which we also accounted for chronological and measurement uncertainty.

The divergence between earlier findings and our own is likely the result of problems with the use of SPDFs as a population proxy[30,31,34]. As discussed above, this approach dubiously conflates process variation—i.e., through-time changes in population level—with the chronological uncertainty inherent in radiocarbon

dates, which has significant analytical consequences for studying population dynamics. As recent simulations studies have shown[31,35], attempting to explain through-time population level fluctuations by comparing this proxy to some covariate (e.g., temperature, another SPDF, etc.) can produce misleading results. Our findings show that this extends to the study of Late Quaternary North American megafauna extinctions, and calls into question the use of SPDFs for studying extinction dynamics.

It is also important to recognise the problems with the radiocarbon record. While North America has some of the most detailed Late Quaternary archaeological and palaeontological records, samples sizes remain limited given the vast spans of space and time involved, and there are a number of important sources of bias (e.g., through-time taphonomic degradation of samples, spatio-temporal sample adequacy, radiocarbon-date calibration artefacts, etc.) that have implications for downstream analyses. Unsurprisingly, debates surrounding the chronology of human arrival to the Americas, founding population size, and subsequent population size fluctuations have continued with no sign of resolution. Concerning the megafauna record, it has long been known that the number of fossil finds in a given region/time to some extent corresponds with archaeological and geological research efforts aimed at dating material thought to be contemporaneous with humans in the Americas[45]. Although we attempted to correct for taphonomic and sampling biases in our analyses (see Methods section), there may still be biases that are difficult to control for at the moment and more research and data are needed.

Nevertheless, our findings make it clear that overkill by rapidly expanding human populations is not supported by the available data. Using the largest assembled database of directly dated North American megafauna, and accounting for chronological uncertainty in the radiocarbon and climate records, our results demonstrate that there is currently no evidence for a persistent through-time relationship between human and megafauna population levels in North America.

Our results are instead compatible with several alternative hypotheses. One is that while climate change appears to have been a dominant driving force behind megafauna population level fluctuations, humans may have been involved in more complex ways than simple overkill models suggest. Indeed, scholars have proposed a number of ways by which humans could have had a significant impact on megafauna populations that do not invoke widespread overhunting and significant population growth. Some

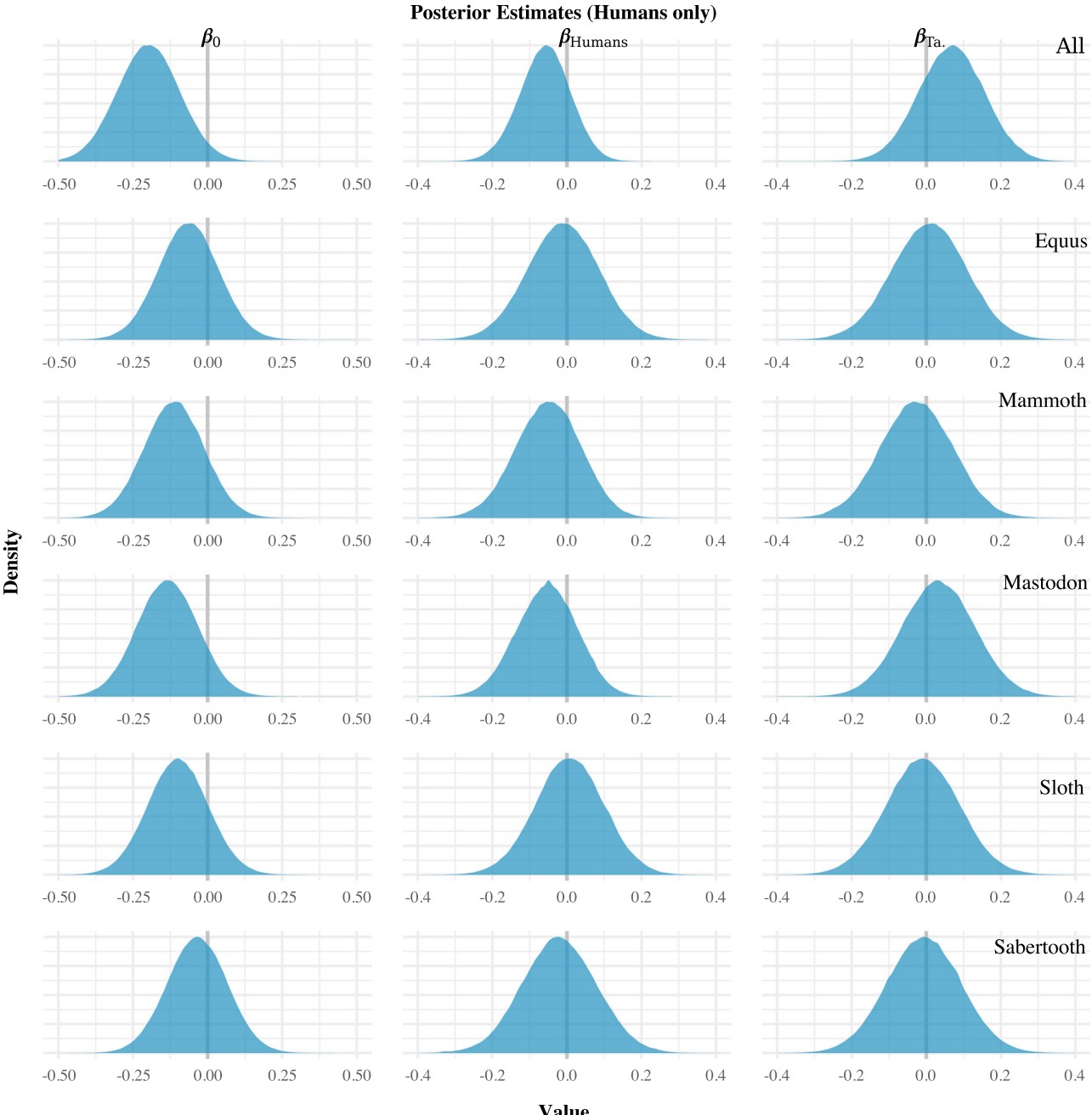

**Fig. 2 Regression results for the "humans-only" models.** The left column ($\beta_0$) indicates the model intercept (the mean level of the megafauna proxy when all other covariates are held constant at zero, which, in this case, has no substantive interpretive meaning); the central column ($\beta_{Humans}$) indicates the effect of human population size on megafauna population size; and the right column ($\beta_{Ta}$) indicates the effect of taphonomy. If human population size was an important driver of megafauna extinction, the estimate(s) would differ (i.e., be non-overlapping) from zero (denoted by the vertical grey lines). Note that in each case, the human model ($\beta_{Human}$) posterior estimates overlap zero, which indicates no relationship between the radiocarbon-date proxies.

have suggested that the depletion of keystone megaherbivores—those animals that have a disproportionately large influence over their environment—led to significant cascading effects on local flora and fauna[46,47], as is known to occur in contemporary ecosystems[48,49]. So, even a few hunters on the landscape targeting only particular species might have led to population declines among numerous megafauna species without any long-term increase in hunting pressure from a growing human population. While this may have been the case for megafauna more broadly, our data indicate that at least some species of megafauna declines occurred prior to declines in keystone megaherbivores. Specifically, final declines in horse and saber-tooth cat population

densities significantly pre-dated those of mammoths and mastodons. In fact, these population declines occurred at a time of increasing mammoth and mastodon numbers, which is particularly interesting in the case of the saber-tooth cat, which is often considered to have been a specialised hunter of these very large animals[50]. Others have proposed that increased competition between humans and carnivores forced carnivores to turn to and intensify predation on other, smaller animals[51,52]. Greater inter-specific competition among carnivores for a smaller and less diverse food source would have driven population declines among not only herbivores but also carnivores, whereas humans may have been able to sustain (or even increase) population sizes

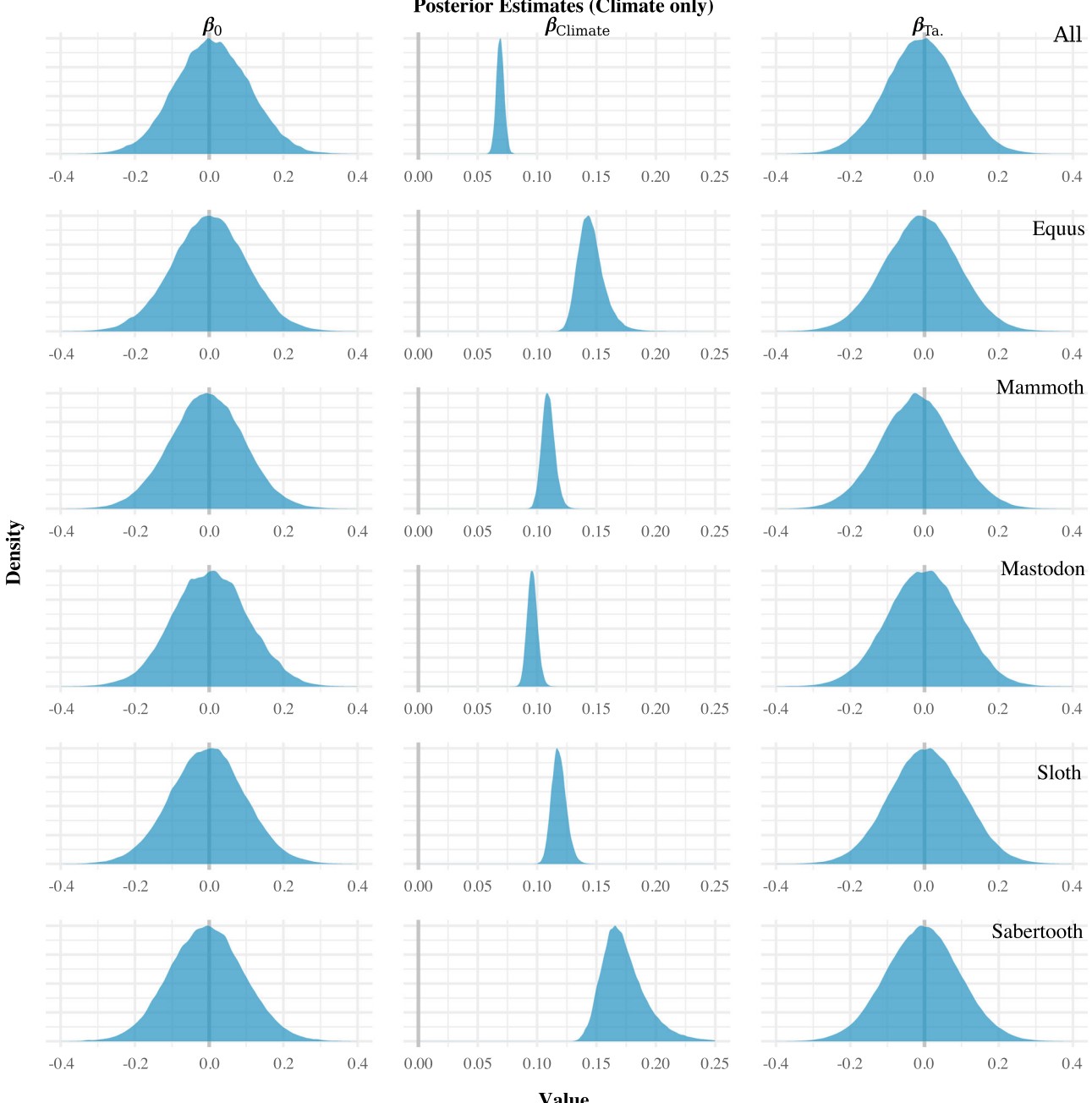

**Fig. 3 Regression results for the climate only models.** Note that the estimates for the climate model parameter ($\beta_{Climate}$) do not overlap zero (denoted by vertical the grey lines).

despite dwindling megafauna numbers by exploiting a broad range of animal and plant foods[50]. Interestingly, there does appear to be a drop in saber-tooth cat population density coinciding with the emergence of Clovis-point wielding peoples in the Americas suggesting that interspecific competition may have had an initial impact on saber-tooth cat populations; although, the sample size for saber-tooth cat is rather small, and the final population decline appears to have occurred closer to the Younger Dryas (YD). Others still have suggested that humans, through hunting and habitat fragmentation, interrupted megafauna subpopulation connectivity, fragmenting populations into smaller, non-viable groups[40,53,54]. Indeed, megafauna, with their large home ranges, small population sizes, and slow life histories are particularly susceptible to extinction by habitat and population fragmentation[46]. If so, the mammoth and mastodon data

suggest that this occurred not with the arrival of Clovis-point wielding people, but much later during the YD.

Alternatively, climate change may have indeed been the primary driving force behind the extinctions, with humans playing no significant role, or perhaps at most performing a coup de grâce on megafauna populations already heading towards extinction. Two key climatic events are often emphasised in the extinction of North American megafauna—the warm Bølling-Allerød interstadial (B-A; ~14.7–12.9 ka) and the cold YD stadial (~12.9–11.7 ka). Indeed, of the 37 genera that went extinct during the late Pleistocene, 16 have last appearance datums (LADs) that fall between 13.8–11.4 ka[55], encapsulating the B-A/YD boundary.

Hypotheses focusing on the B-A assert that rapid temperature increase and associated ecological changes led to the extinction of North American megafauna. Some scholars have argued that the

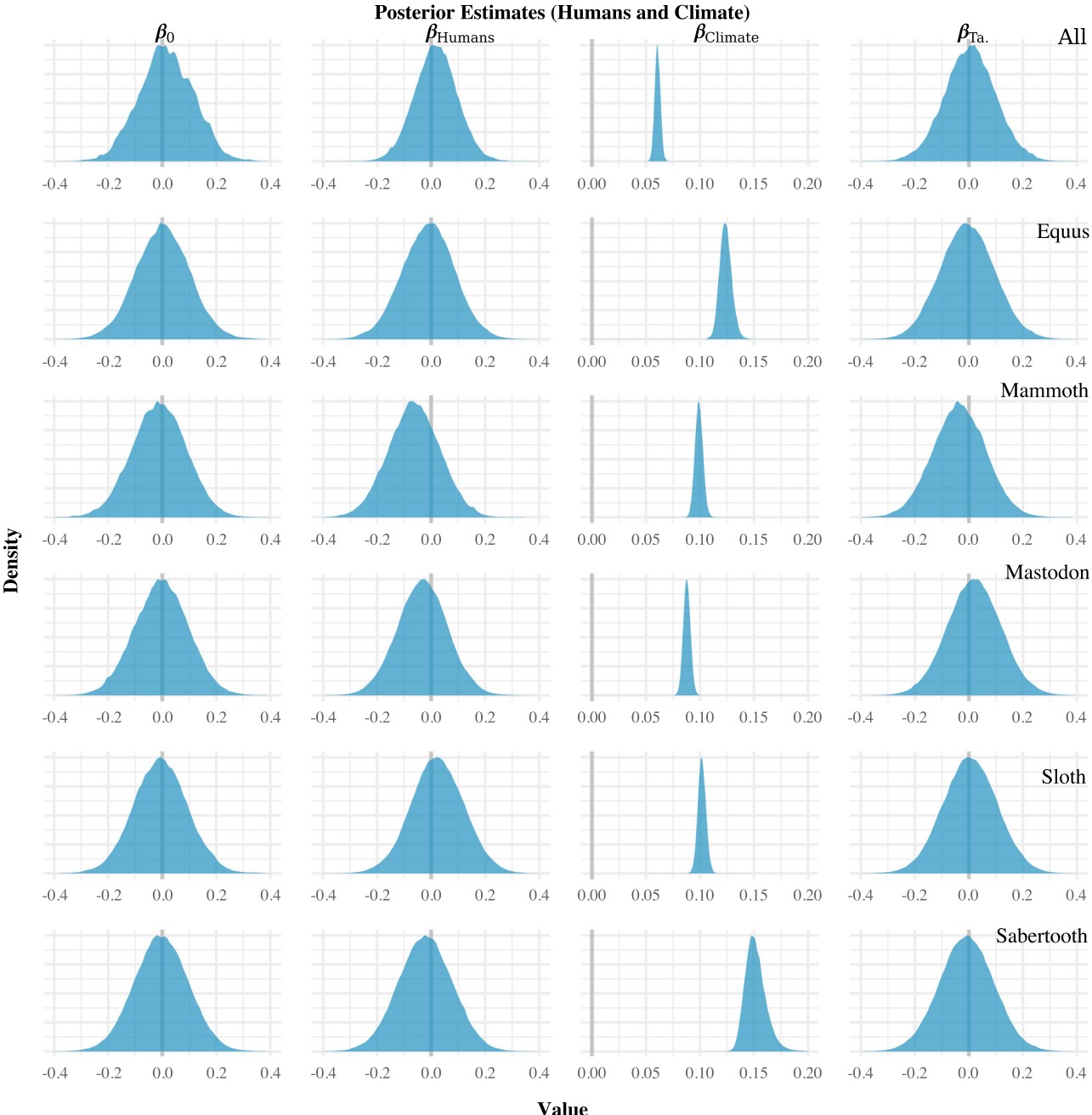

**Fig. 4 Regression results for the mixed human and climate models.** Note that for humans ($\beta_{Humans}$) the estimates are centred on zero, whereas for climate ($\beta_{Climate}$) the estimates do not overlap zero (denoted by the vertical grey lines).

abrupt warming associated with interstadials drove megafauna extinctions across the Americas and Eurasia[40,41]. In North America, for instance, Cooper and colleagues[40] posited that megafauna extinctions corresponded with or closely followed the abrupt warming of the B-A, and similarly timed megafauna population declines have been inferred from declines in *Sporormiella* spore abundance, a fungus found in the dung of ungulates and used as a proxy for megafauna population level changes[56,57]. Even though our analysis identified a positive relationship between temperature and megafauna population levels, it's possible that a non-linear relationship exists. Megafauna populations levels could, for example, have generally increased along with temperature, giving rise to the relationship we identified, but then crashed in response to rapid temperature shocks (like the onset of the B-A) that crested some currently

unidentified eco-biological threshold. Further, the extreme temperature upturns that characterise the Pleistocene interstadials are typically followed by more gradual temperature downturns. This means that a rapid increase may trigger ecosystem changes that precipitate population declines, which, in turn, correspond to subsequent temperature declines giving rise to the positive correlation between the climate and population proxies. Currently, the fossil evidence is not sufficient to evaluate this non-linear model, and our analysis cannot rule it out.

That said, we think our findings point more clearly to the onset of the YD as driving megafauna declines and extinctions. As can be seen in Fig. 5, the density of megafauna dated events increases during the B-A before decreasing shortly after the onset of the YD. The same can be said independently for mastodon, sabertooth, and sloth populations, whereas final mammoth population

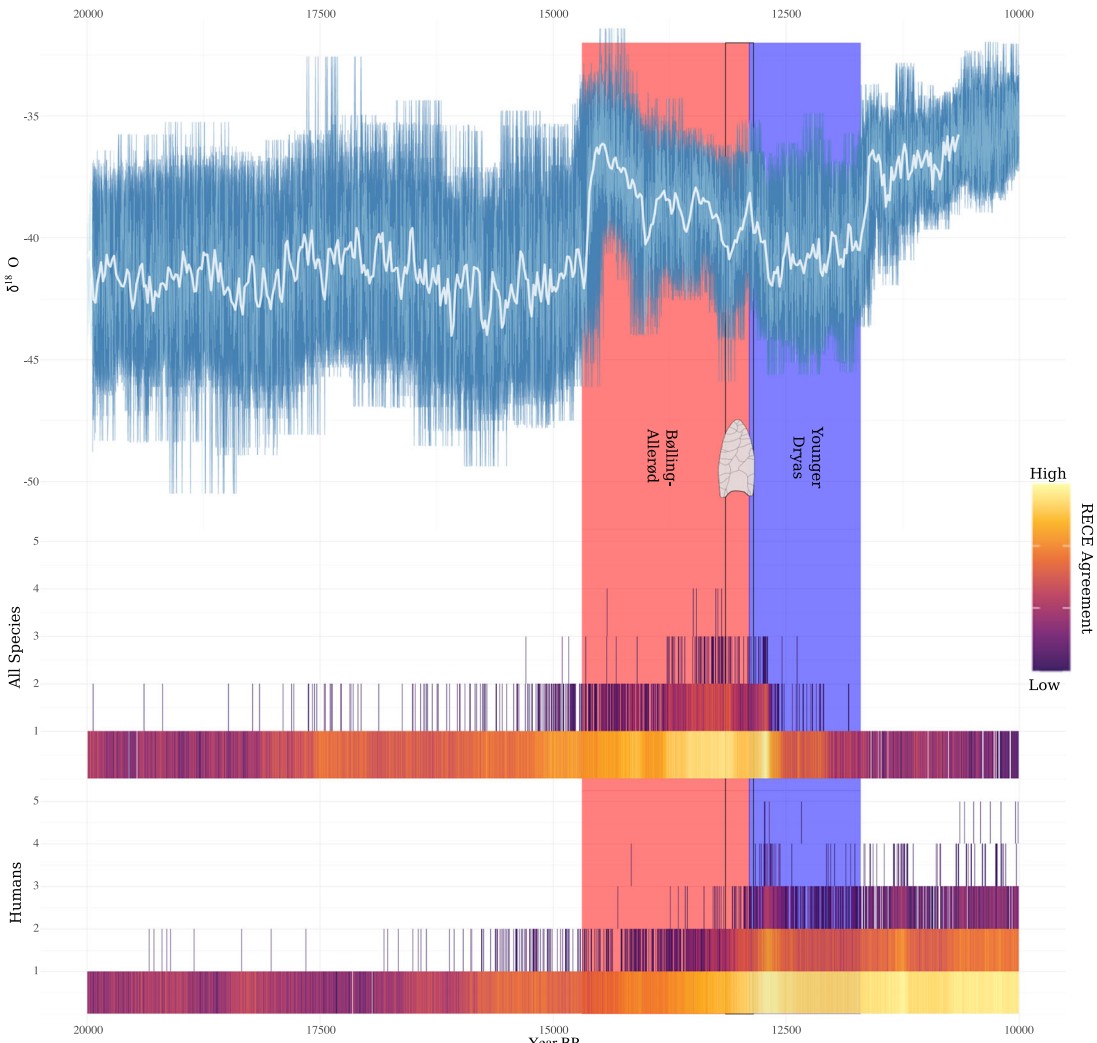

**Fig. 5 Climate, megafauna population, and human population changes through time.** Annual NGRIP oxygen isotope (δ18O) record with temporal uncertainty re-projected into the measurement domain (top). The white line represents the smoothed (50-year running mean) NGRIP record used in previous research and the main analyses of our study. The light blue line represents the 95% uncertainty envelope created by re-projecting temporal uncertainty from the time domain (x-axis) onto the measurement domain (y-axis) using Boers et al's[42] approach. Radiocarbon-dated Event Count Ensembles (RECEs) for megafauna and humans (bottom). RECEs are plotted in colourmap (magma) so that higher (brighter colours) and lower (darker colours) density regions can be visually distinguished. The y-axis represents the count—a count of two, for example, would result from two dated events occurring in the same year in a RECE member. The heat-map colour scale is plotted on the right of the figure and the numbers (agreement) values have been logarithmically stretched in order to enhance contrast. Clovis period delineated by the black box. See the 'how to read a RECE' Supplementary Fig. 9 for help with interpreting RECE plots.

declines appear to have occurred later in the YD, and final horse population declines appear to have occurred during the terminal B-A (Fig. 7).

While some scholars have dismissed the YD as a driver of megafauna extinctions, arguing that conditions were no more severe than earlier cold periods through which North American megafauna survived (e.g., the Last Glacial Maximum [LGM])[1], there is evidence to suggest that the YD involved a specific set of climatic and ecological changes that may have been particularly devastating to megafauna populations[58]. During the YD, summer insolation (seasonality) reached one of its highest peaks of the late Pleistocene[59], atmospheric $CO_2$ concentrations rapidly rose[60], and some of the fastest vegetation changes of the Late Glacial occurred[61,62]. Increased seasonality would have brought about shorter peak plant growth seasons, declines in plant community heterogeneity, and shifts in plant anti-herbivory defences[63,64]. Consistent with the fossil record, these conditions favoured smaller-bodied ruminants (e.g., moose, white-tailed deer) which can subsist

on less diverse diets and are better equipped for exploiting plant nutrients across shorter growing seasons than larger-bodied monogastric animals and/or those with slow life histories (e.g., mammoth, mastodon, sloth)[64]. Increased atmospheric $CO_2$, which is known to reduce plant nitrogen content, may have led to poorer quality forage and reduced landscape carrying capacities[65]. And in some regions, major vegetation changes in response to climate change were so rapid (<100 years)[59,61,62] that megafauna populations may not have had time to adapt. Indeed, some studies have found declines in animal and plant biodiversity[66,67] and genetic diversity[37] occurring around the onset of the YD which have been tied to climate change and, occasionally, to more catastrophic events such as an extra-terrestrial impact[68], although the latter are often disputed[69].

The conditions of the YD were, however, not felt evenly across North America[58], and, unsurprisingly, animal responses to these changes varied across both time and space e.g. refs. [24,37,70]. On that note, it is important to consider the evidence divided into

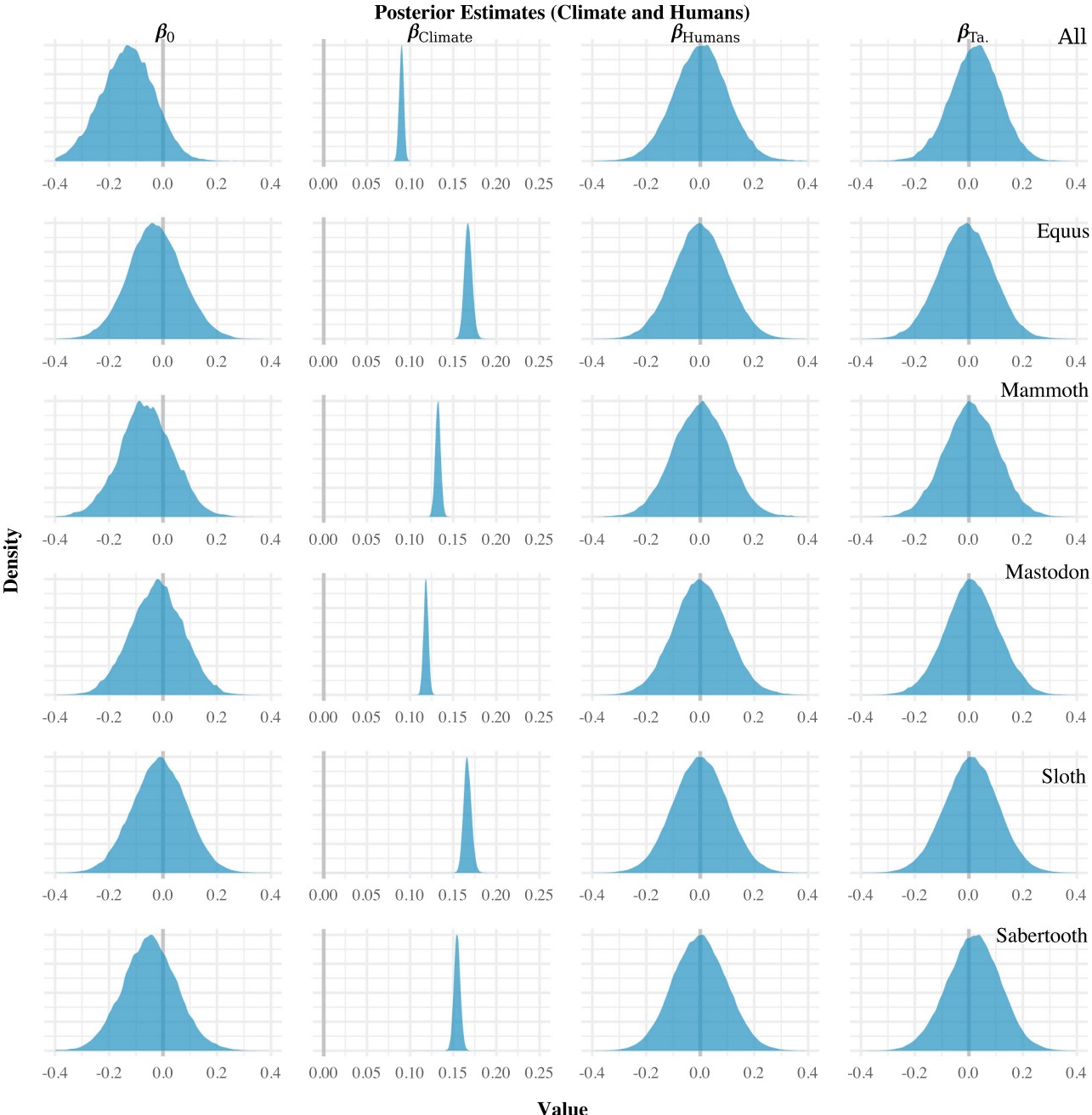

**Fig. 6 Regression results of the extended climate analysis.** Consistent with our earlier findings, the climate ($\beta_{Climate}$) estimates do not overlap zero (denoted by the vertical grey lines), whereas the human ($\beta_{Human}$) estimates centre on zero.

the Southwest and Great Lakes regions, drawing on the available local palaeoclimatological and palaeoecological records and our results.

In the Great Lakes region, mastodon and mammoth show an overall increase in dated event counts during the B-A and a decrease coinciding with the onset of the YD or shortly afterwards (Fig. 8). Pollen, stable isotope, and lake level data indicate cooler (~5 °C) and drier conditions in the Great Lakes region during the YD[71]. Vegetation changes were rapid, both in terms of their abruptness (within a century of the YD onset) and rate at which vegetation expanded geographically (>300 km/century)[59]. While a number of scholars have argued that megafauna declines significantly pre-dated major plant community changes in the Great Lakes region[56,57,72], our findings suggest that YD climate, megafauna population, and plant community changes closely

tracked one another. Open spruce (*Picea*) and sedge dominated parklands were quickly replaced with westward migrating mixed pine (*Pinus*) forests, with corresponding increases in taxa such as birch (*Betula*), elm (*Ulmus*), and oak (*Quercus*)[56,57,59,61,72]. The reduction of open environments in the case of mammoth, and of spruce woodlands in the case of mastodon, may explain the extirpation of these megafauna from the region[73].

Likewise, in the American Southwest, sloth and mammoth show an overall increase in dated event counts during the B-A and a decrease around the onset of the YD (Fig. 9). That megafauna populations seemingly increased during the B-A is interesting given that speleothem and paleolake records indicate that the region experienced significant aridification[74–76], being described by some as the worst drought in the American Southwest in 46,000 years[77]. Most speleothem and lacustrine

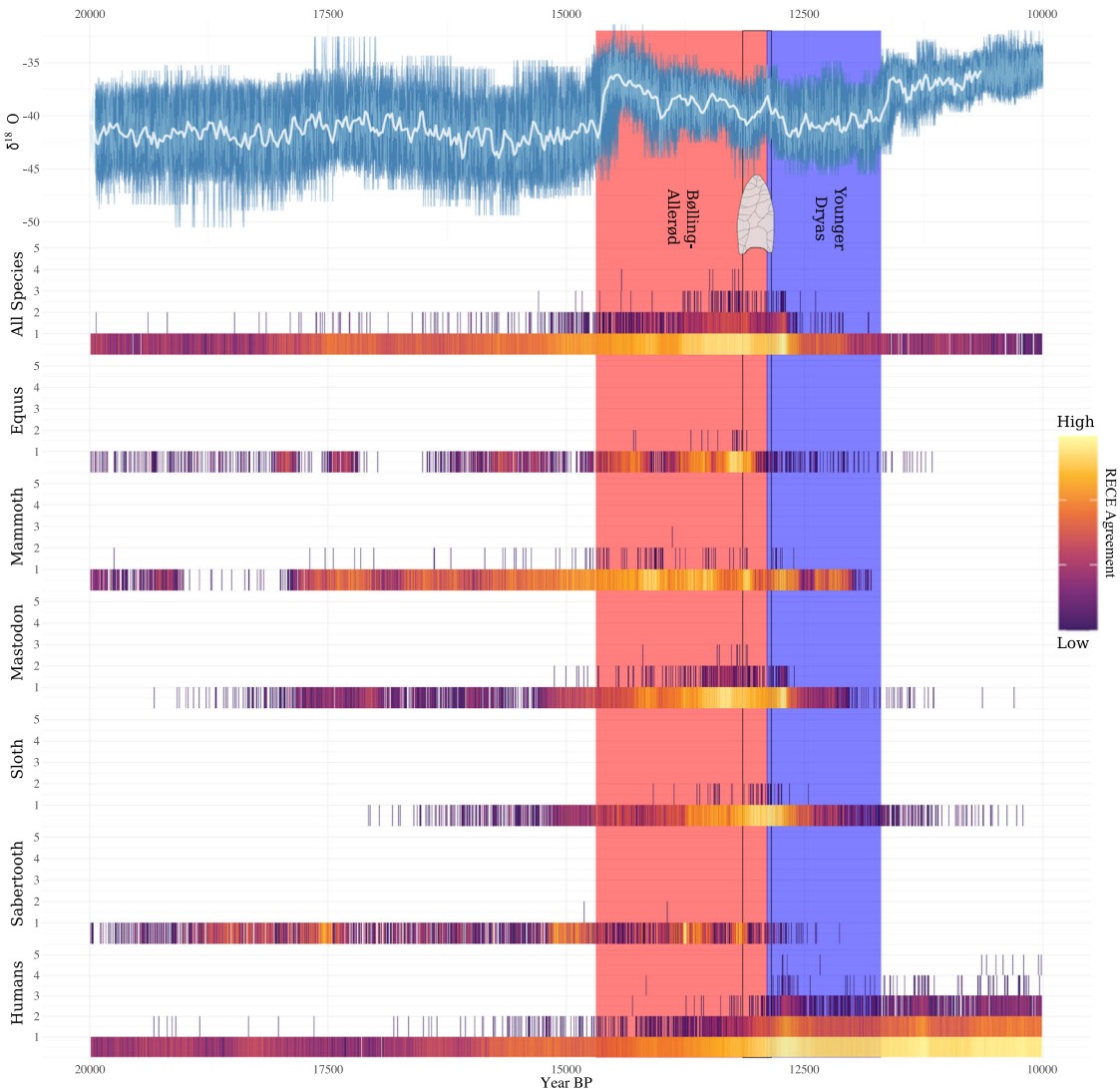

**Fig. 7 Climate, human and megafauna population changes through time.** Annual NGRIP oxygen isotope (δ[18]O) record with temporal uncertainty re-projected into the measurement domain (top) and Radiocarbon-dated Event Count Ensembles (RECEs) for humans, megafauna, and megafauna broken down by taxa (bottom). For more details see Fig. 5.

records indicate cooler and wetter conditions following the onset of the YD e.g. refs. [74,75,77–79], although it has been suggested that the Southern High Plains experienced some periods of drought[80]. Plant community responses to these climate changes appear to have been highly variable and/or poorly resolved and "consensus does not exist regarding the magnitude of direction of YD climate change in the Southwest" (Ballenger et al.[81], p. 511). For instance, some plant pollen and macrofossil records indicate a transition from open environments to woodlands dominated by pine, oak, and juniper[81,82], others indicate a somewhat opposite trend[83], and others still show little change across the B-A/YD transition[84]. Attempting, then, to explain the American Southwest megafauna population declines as a response to changes in plant communities is, at present, difficult.

In summary, the results of our quantitative analyses are consistent with climate-driven declines in North America's megafauna populations. Data quality issues aside (see Introduction), using the largest assembled database of directly dated megafauna, we found no through-time relationship between megafauna and human population levels. While this does not preclude humans from having had an impact—for example, by interrupting megafauna subpopulation connectivity or performing a coup de grâce on

already impoverished megafauna populations—it does suggest that growing populations of "big-game" hunters were not the primary driving force behind megafauna declines and extinctions. Instead, we found a consistent positive correlation between megafauna population levels and the NGRIP climate proxy. In other words, decreases in global temperature correlate with decreases in megafauna population levels. Final megafauna population declines leading to extinction roughly coincided with the onset of the YD, hinting that the unique conditions of the YD—i.e., abrupt cooling, increased seasonality, increased $CO_2$, and major vegetation changes—played an important role in the extinction of North America's megafauna. Furthermore, these findings suggest that YD climate, megafauna population, and plant community changes were in approximate equilibrium.

The causes(s) of North American Late Quaternary megafauna extinctions are likely to remain contentious. While to many researchers it may be an unlikely coincidence that megafauna extinctions coincided with human arrival at different times and in different parts of the world, it remains important to scientifically demonstrate this. And in doing so the limitations of the record are readily apparent: we simply do not have robust records for fauna and humans for vast spans of time and space. Building

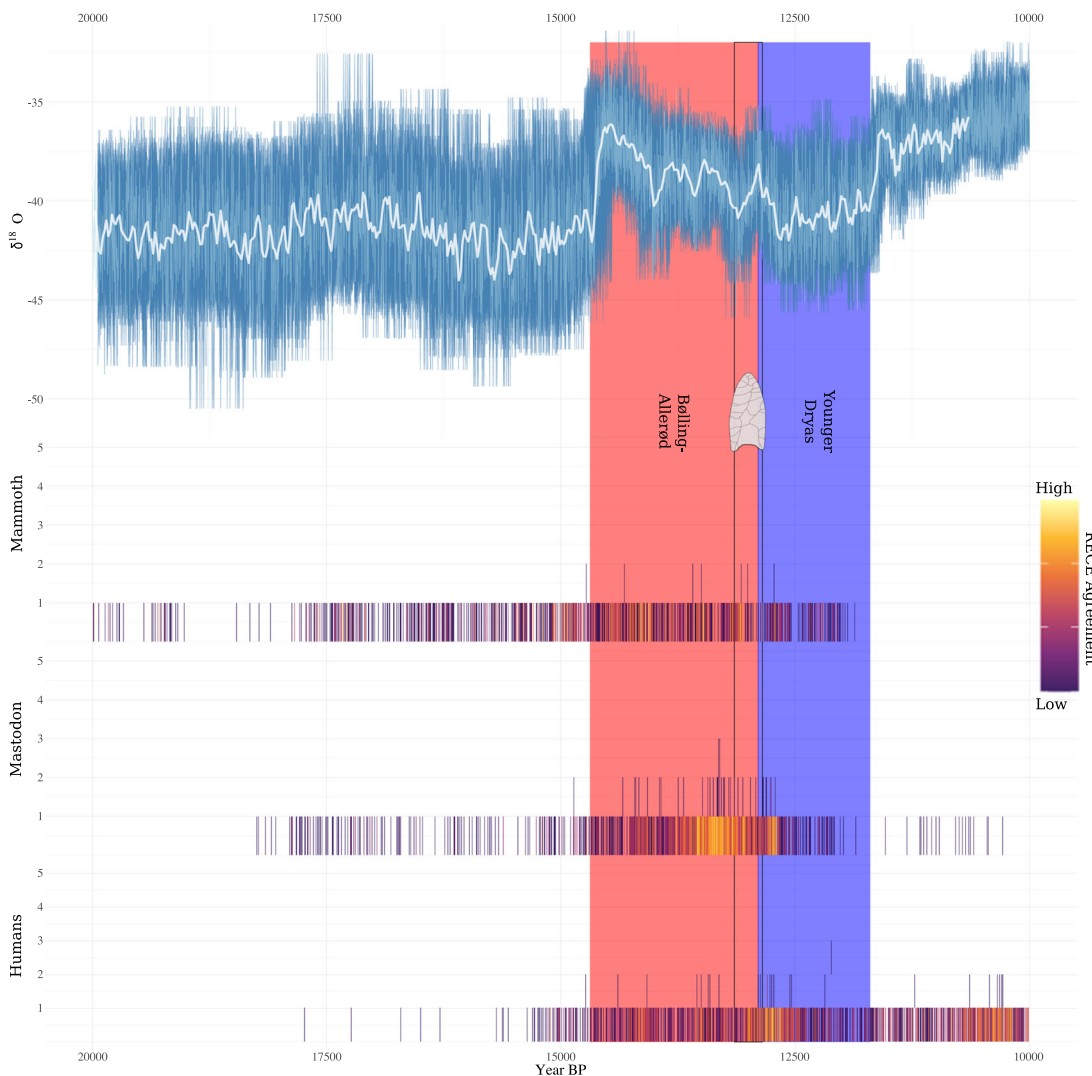

**Fig. 8 Climate, mammoth population and mastodon population changes through time in the Great Lakes (GL) region.** Annual NGRIP oxygen isotope (δ¹⁸O) record with temporal uncertainty re-projected into the measurement domain (top) and Radiocarbon-dated Event Count Ensembles (RECEs) for mammoth and mastodon for the Great Lakes region (bottom). For more details see Fig. 5 caption.

reliable records, and developing robust methods for interpreting them, remains a key task.

## Methods

**The data.** The megafauna radiocarbon database used in the present study was compiled by Broughton and Weitzel[24] (Supplementary Data 1). It is the largest yet assembled for North American megafauna, comprising 521 radiocarbon-dated individual megafauna from the contiguous US and a small sample from immediately adjacent regions of Canada (n = 14). Only direct AMS or conventional radiocarbon assays were included. Dates derived from apatite, whole bone, or bone mineral, and those from unequivocal archaeological contexts (i.e., sites with clear kill and/or scavenge associations), were excluded (for a more detailed description of the data collection protocols see Broughton and Weitzel[24]).

We noticed that there may exist a number of instances where multiple dates originate from a single individual or event (i.e., a single defecation event). Since repeated samples of the same event would lead to over counting for some events, we collated an additional "chronologically cleaned" dataset (Supplementary Data 2). As a first step, we flagged instances where dated fossils of the same taxon from the same locality overlapped in their ages (i.e., radiocarbon years before present [RCYBP] ± error). As an example, two *Equus* remains at Paisley Cave 5 (Lab No: UCIAMS-103087 & UCIAMS-103088) returned overlapping ages of 11770–11850 and 11780–11850 RCYBP. Relevant literature was then consulted to see whether it was possible to attribute remains with overlapping dates to separate individuals, say, on zooarchaeological or stratigraphic grounds. We then generated a dataset with all non-separable overlapping dates flagged, and these were removed prior to analysis. Given the large size of the Rancho la Brea site and fossil assemblage, and efforts to target individual megafauna and specific skeletal

elements (femoral shafts, for instance) e.g. refs. [85,86], we opted to include all dates from this site under the assumption that most originate from unique individuals. All dates taken from disaggregated plant remains at Bechan Cave were excluded. The resulting database comprised 432 radiocarbon dates and these data were used in our supplementary analysis. We reasoned that any major differences in findings between our analysis involving the whole dataset and the analysis involving the filtered one would have indicated that potentially important biases might have been present in the data and that a more nuanced approach to event identification would be necessary e.g.,[36]. The supplementary analysis results, however, were consistent with our other findings.

The archaeological radiocarbon database, limited to 15–10 ka, was gathered from the Canadian Archaeological Radiocarbon Database (CARD). Given that the CARD database comprises minimally-vetted, user-supplied radiocarbon data, Broughton and Weitzel[24] cleaned the data by removing: (1) duplicates; (2) data marked as "anomalous"; (3) dates derived from non-anthropogenic contexts (i.e., those derived from palaeobiological and geological contexts); and (4) dates derived from megafauna skeletal material but lacking clear evidence for a kill/scavenging association. This screening served two key purposes. Firstly, criteria 1 & 2 cleaned the dataset for data quality—for example, by removing dates obtained through non-modern dating methods; and secondly, criteria 3 & 4 ensured that the dated being analysed could be confidently associated with the archaeological record. This vetting produced a database of 938 anthropogenic radiocarbon dates and was provided to us by the authors. The vast majority of sites are evenly represented in respects to their radiocarbon record—i.e., most sites in our analysis are represented by a single radiocarbon-date (Supplementary Fig. 10).

As a proxy for climate change (temperature) we used the ~50-year resolved North Greenland Ice Core Project (NGRIP) δ¹⁸O record, whereby a rise in δ¹⁸O‰ is considered to represent a rise in air surface temperature[43] (Supplementary

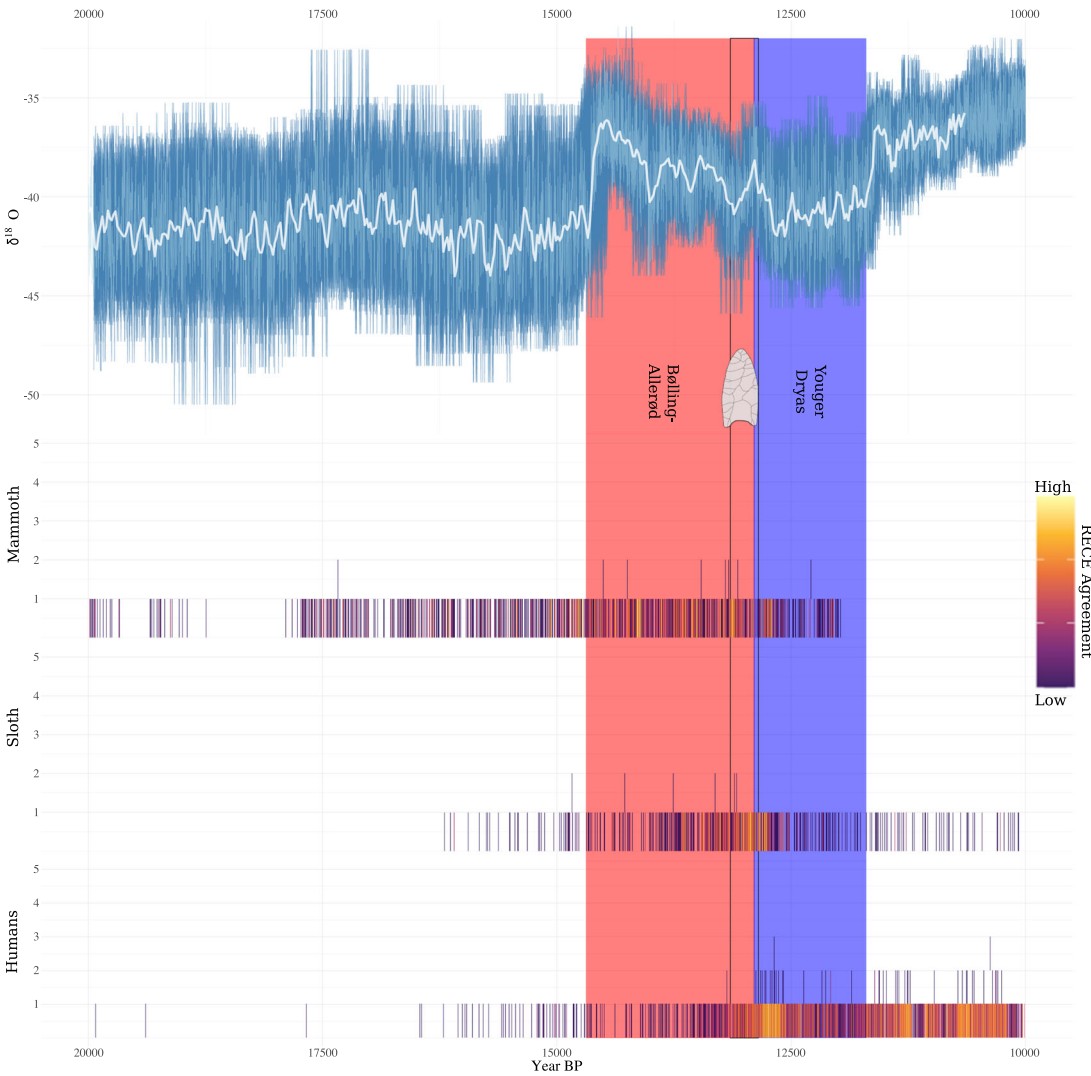

**Fig. 9 Climate, mammoth population and sloth population changes through time in the American Southwest (SW).** Annual NGRIP oxygen isotope (δ[18]O) record with temporal uncertainty re-projected into the measurement domain (top) and Radiocarbon-dated Event Count Ensembles (RECEs) for mammoth and sloth for the Southwest region (bottom). For more details see Fig. 5 caption.

Data 3). The NGRIP record matches well with other highly-resolved temperature proxies from North America and Western Europe[75,87,88], and, as such, is considered a suitable proxy for broad-scale long-term trends in late Pleistocene Northern Hemispheric temperature fluctuations.

**Radiocarbon-dated event count model**. REC models are a novel regression approach designed to account for chronological uncertainty in REC time-series[34]. In the present case, "event" effectively refers to the death of an animal, and a corresponding count-based time-series (event count sequence) would ideally indicate through-time changes in the number of animal deaths over a given period. The events in question, however, are dated with radiocarbon assays. These date estimates have uncertainties associated with them, which are represented by distributions of possible dates that can span many centuries. A single event (animal death) might have occurred at any time within the domain (timespan) radiocarbon-date distribution associated with the relevant fossil. Thus, the true time-series of fossil counts cannot be established because individual fossils cannot be assigned to any given date with absolute certainty. Multiple potential event count sequences are, therefore, always possible. In other words, REC models explore alternative scenarios consistent with probabilistic distributions of chronological data. Given chronological uncertainty, the sequence of dated events and the through-time relationship between different categories of events is, by no means, self-evident.

REC models account for chronological uncertainty by employing a Bayesian hierarchical approach to regression modelling[34]. This is accomplished by (1) sampling different probable event count sequences, (2) using each one as the dependent variable in a separate regression, and then (3) nesting the models in a Bayesian multi-level framework with priors for the main regression parameters.

The sample of alternate probable event count sequences, called a Radiocarbon-dated Event Count Ensemble (RECE), is effectively a set of 'what-if' scenarios. Each individual sequence in the RECE (a single member of the ensemble) represents one of the probable event count sequences that might have occurred in the past. Likewise, the corresponding regression models represent a sample of probable regressions. The estimated parameters for these individual regressions (e.g., mean regression coefficients) can be thought of as having been drawn (sampled) from hyper-parameter distributions—where a "hyper-parameter" is a parameter of a distribution that represents the uncertainty in another parameter (the mean of a distribution of other means, for example). So, at the top-level, a REC model has one or more hyper-parameters that define the parent distributions of lower-level parameters estimated for each of the individual regression models. In our analysis, for instance, one of the hyper-parameters is an estimate of the mean of the sample of regression coefficients pertaining to the human population covariate. It captures the variation between individual regressions with respect to the relationship between human population sizes and megafauna population sizes. Importantly, this variation is caused by the differences between possible event-count sequences. REC model hyper-parameter distributions, therefore, reflect the impact of chronological uncertainty on the regression analysis without confusing that uncertainty with process variation (through-time changes in event counts).

To begin, we produced a RECE for the megafauna fossils. Each RECE member was constructed by randomly sampling a probable date from each radiocarbon-date density in our database—we sampled from the 98% confidence region of each density in accordance with the level of the density function. This sampling process results in a set of probable fossil ages (dates in years BP), one for each fossil in the database. Then, the dates were counted to produce one probable sequence of fossil counts. This sampling process was repeated to compile the ensemble of alternate probable sequences (i.e., the RECE).

Next, each member of the RECE was used in an appropriately specified count-based regression model. Following Carleton[34], we used a Negative-Binomial (NB)-REC model. One of the key reasons for choosing this distribution is that it is appropriate for count data, but more importantly it can account for temporal spread, which is a known effect of chronological uncertainty[33]. In the present study, we extended the NB-REC model further to account for chronological uncertainty in the covariates. The human population size proxy, for instance, is the same as the megafauna one, namely a REC proxy. This means that we could straightforwardly adapt the model to include samples from the RECE of the human data. Using similar logic, we also accounted for chronological uncertainty in the taphonomic proxy and the NGRIP data. For more information on model choice and parameters see Supplementary Note 3.

**Regression models**. We used our extended NB-REC regression models to test for relationships between megafauna population size and two potential explanatory variables, namely human population size and climate change. Following and extending the analytical setup used by Broughton and Weitzel[24], we ran several parallel analyses. Each one involved three models: (1) one in which the megafauna data were compared to a radiocarbon-date database comprised of human or anthropogenic samples (i.e., a human population proxy); (2) one in which the megafauna data were compared to the NGRIP oxygen isotope record (i.e., a climate change proxy); and (3) one model in which megafauna data were compared to both the human population proxy and the climate change proxy. In one of these three-model analyses, we used the whole megafauna radiocarbon-date database as the response variable for the regression models. Then, for the other analyses, we subdivided the megafauna radiocarbon-date database into the same five taxa groupings used by Broughton and Weitzel[24]. These species sub-samples were slotted in as the regression response variable for each of the three models in a given analysis. This meant that in addition to the analysis involving the whole megafauna radiocarbon-date database, we ran analyses that looked separately at horse (*Equus* sp.), mammoth (*Mammuthus* sp.), mastodon (*Mammut americanum*), Shasta's ground sloth (*Nothrotheriops shastensis*), and saber-tooth cat (*Smilodon fatalis*) count data. We also ran a further two analyses based on Broughton and Weitzel's[24] regional subdivisions. One involved samples related to mastodon and mammoth found in the Great Lakes region, whereas the other involved samples related to sloth and mammoth found in the US Southwest. In each of these regional analyses, a separate set of models was created for the relevant megafauna taxa. It is important to note that we also used the same temporal divisions as Broughton and Weitzel[24] and, therefore, our models were restricted to the period from 15.0–11.7 ka.

To account for the taphonomic loss of older samples from the archaeological and paleoenvironmental records, we used a proxy for taphonomic processes recommended by Surovell and colleagues[39] as a control variable in all of our regression models (Supplementary Data 4). This proxy is based on a northern hemispheric tephra record and can be interpreted as an indicator of the loss of evidence over time caused by taphonomy. Crucially, this proxy would be subject to any time-dependent and climate-dependent process that affect taphonomy on a regional scale. By including it as a covariate in our regressions, we allowed for the possibility that these taphonomic processes account for through-time variation in fossil counts. If the relevant regression coefficient was determined to be non-zero, then the taphonomic proxy would explain some of the variation in fossil counts thereby reducing the variation leftover for other covariates to explain. Alternatively, if the taphonomic proxy regression coefficient was estimated to be zero, it would indicate that regional, long-term taphonomic processes cannot explain variation in fossil counts. As with the other proxies, though, the taphonomic record contains chronological uncertainty. Therefore, to account for that uncertainty, we followed the same procedure we used for the megafauna and archaeological data. We sampled probable tephra event count sequences, and included each sampled sequence in one of the probable regressions as a covariate, along with a probable human event count record and/or the NGRIP climate change proxy.

To determine whether megafauna population declines were likely related to human population increase, climate change, or both, we examined the posterior mean densities of the regression coefficients in the NB-REC models. We reasoned that if human population size, climate change, or both were important drivers of Late Quaternary megafauna extinctions in North America, then the regression coefficient(s) associated with the relevant variable(s) would be significantly different from zero. More specifically, a given variable would be considered important if its corresponding posterior density estimate did not include zero within its 95% credible region. Since we included the taphonomic proxy in all models, any significant effects could be viewed as indications that a given variable was important even after accounting for trends in megafauna population levels caused by taphonomic processes.

With such an enormous number of model parameters to estimate (see Supplementary Note 3), computational resources were a limiting factor. Therefore, we decided to sub-sample the data to make computation feasible. We employed two sub-sampling strategies. First, we used RECE samples comprised of 50 probable event count sequences and, second, we sub-sampled the RECEs with respect to time so that only every 10th observation was included. The former, of course, means that we cannot fully explore the tails of the target parameter value densities because we have likely not included enough of the variation in probable

event count sequences to fully capture the chronological uncertainty in the relevant records. That said, some accounting of that uncertainty is, in our view, better than none. The second sub-sampling strategy meant that we examined every 10th year from the beginning of a given RECE's temporal span to the end of its span. Previous simulation work investigating REC models has shown that this sub-sampling has no obvious effect[34], but it should be noted that it clearly would have an impact if the process of interest had important high-frequency variation. It would also be a problem if the sub-sampling was too aggressive, involving inter-observation gaps so large that important patterns in the records were obscured. In the case of the megafaunal extinction question, however, even decadal sampling (as was done here) is a very high resolution relative to the millennial scale processes under investigation.

All analyses were conducted in R[89] using the Nimble package[90] for estimating Bayesian model parameters with MCMC. We then used "ggplot2"[91] and "ggpubr"[92] for plotting results and diagnostics. A compiled R-markdown pdf is provided as Supplementary Code 1. The uncompiled R-markdown code file is available at https://github.com/wccarleton/megafauna-na.

**Reporting summary**. Further information on research design is available in the Nature Research Reporting Summary linked to this article.

## Data availability
All data generated or analysed during this study are included in this published article (and its Supplementary Data files) with the exception of the human radiocarbon sample database, which can be provided on request or retrieved directly from CARD, but cannot be openly shared according to the policies of the source database (https://www.canadianarchaeology.ca/)

## Code availability
All analyses were conducted in R (4.03) using a combination of established packages and custom code. A zip file containing data an R markdown file has been uploaded as Supplementary Code 1. The R markdown file contains a thorough description of the analytical process including documented code for ease of replication. The following R packages were used in the analysis and figure production: nimble (0.9.1), ggplot2 (3.3.2), ggpubr (0.4.0), clam (2.3.5), tidyr (1.1.2) and abind (1.4.5).

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

## Acknowledgements

We thank Jack Broughton and Elic Weitzel for the often-thankless task of compiling radiocarbon databases and for providing us with their data over email. We acknowledge the Max Planck Society for funding.

## Author contributions

M.S., W.C.C. and H.S.G. conceived the project. M.S. and W.C.C. collated the data. W.C.C. ran the statistical analysis. M.S., W.C.C. and H.S.G. wrote the manuscript. All authors contributed to the manuscript and approved the final version.

## Funding

## Competing interests

The authors declare no competing interests.
