## [Peer Review File · Nature Communications]

Reviewers' Comments:

Reviewer #1:

Remarks to the Author:

This is an excellent manuscript. It is a thoughtful and well-executed study that contributes to our understanding the timing and cause(s) of the extinction of North American Megafauna –a topic of critical importance to researchers focused on the Late Quaternary. In addition, this paper's critique of use of summed probability density functions (SPDF) and their application of regression analysis with Radiocarbon-dated event count (REC) modeling means this paper has the potential to make significant methodological contributions to numerous fields of study. Its overall conclusion that the decline in North American megafauna is best explained by a changing climate, rather than an increase in human populations, is reasonable and generally fits other lines of evidence which are well- summarize in the introduction and discussion sections of this paper. Readers of Nature Communications will likely be highly interested in this paper. This said, there are a few relatively minor revisions to its data presentation and its evaluation of its key findings that should be made before final publication.

The use of Radiocarbon Event Count Ensembles (RECEs) represents a useful new analytical tool. However, I feel the description of this analysis and how these data should interpreted needs more clarification, especially on page 4 (lines 14-19) and page 17 (as well as in the supplementary information). Including the 2020 JQS article in the supplemental material helped in understanding of this technique. But the average reader will benefit from more discussion of this procedure in the main paper. The authors should also more carefully explain the interpretations of Figures 5, 7, and 8 (and similar graphics in the supplementary data) where RECE counts are presented. I do not think it is entirely clear to the reader how to identify the "higher density regions (darker areas)". In addition to a better description of this analysis, these figures might also be improved if a gradation of colors was used (such as "heat maps") to depict the RECE rather than just using black and grays. The RECE lines are particularly hard to see in the less dense areas or when counts rise to 2 or 3.

In addition, while results of the regression analyses (e.g., figure 2-4 & 6) are pretty straightforward, I did wonder whether the reader should place any importance on the fact that the regression coefficient (β_{climate}) values for Sabertooth and Equus appear to be significantly higher than the other taxa? Based on Figure S9, Sabertooth and Equus appear to go extinct earlier than the other taxa. I do note that the brief discussion on page 14 (lines 34-39) seems to present a slightly different interpretation of Figure S9—which possibly relates to my comments above. In any case, if we accept the conclusion that climate change is the predominate variability influencing megafauna extinctions should there not be more consideration of the nature of the climate change at the time of these species disappear?

Overall, the discussion section of this paper is thorough and thoughtful. However, I believe the authors have data that could be used to explore some of the alternative hypotheses more deeply. For example, on page 13 (lines 35-51), there is a brief discussion of how the loss of keystone megaherbivores and interspecific competition among carnivores and humans may contribute to megafauna extinction in the absence of increasing human populations. This paper has data (for example figure S9) which seemingly could be leveraged to consider these alternatives in more detail. Mammoths and Mastodons are widely discussed to be keystone species, so what is the implication of them surviving longer than the other species considered in this paper? Likewise, if Sabertooth disappeared centuries before evidence for increase human populations does that have any implications for interspecific competition?

Finally, the authors choice to use Broughton and Weitzel dataset for North American 521 megafauna is reasonable and justifiable given its size and quality. Further, the authors use of a modified subset of just 432 specimens also is reasonable given their concern about duplication of samples. However, the supplementary data they provide appears to be simply a copy of the Broughton and Weitzel's original data. I would suggest adding a column in this file which identifies the samples dropped from consideration, or perhaps simply limit the table to just the specimens included in the analysis. It also

would be simple for add that the authors to add 20 or so additional Mastodon dates, which have been recently published in Karpinski, E., Hackenberger, D., Zazula, G. et al American mastodon mitochondrial genomes suggest multiple dispersal events in response to Pleistocene climate oscillations. Nature Communications 11, 4048 (2020).

Reviewer #2:

Remarks to the Author:

NCOMMS-20-39106 – Climate change, not human population growth, correlates with Late Quaternary megafauna declines in America

Reviewer's statement:

Cutting straight to the chase, I have no hesitation in recommending that this paper is published in the esteemed Nature Comms journal. For the Editor and the Authors, to place my few comments in context, I have a background in using Bayesian statistics to study Late Glacial and Mesolithic radiocarbon datasets obtained from archaeological sites to reconstruct human population demographics in northern Britain. As such, some of the territory covered by this paper is unfamiliar study ground in terms of its geographical range and faunal histories. That said, I hope my few comments below will provoke some thought concerning alternative approaches to analysing radiocarbon datasets. They are by no means posited to stymie the robust statistical analysis undertaken by the authors.

This is a stand-out paper providing an interpretation of radiocarbon data using Bayesian statistical methods to show that megafaunal extinctions in North America are chronologically aligned with climate change events occurring at the Late Pleistocene-Early Holocene transition. Its findings challenge alternative hypotheses attributed to human-related impacts.

This paper will be of significance to a wide community of archaeological prehistorians, ecologists and palaeogeographers with interests in climate change, extinction events and human demographic studies ranging across the Americas, Eurasia and Antipodean continents. The authors have taken a novel approach not seen frequently (if at all) in similar studies, by using a Bayesian statistical analysis – that of so-called Radiocarbon Dated Event-Count (REC) Modelling.

I recommend that the authors expand the rationale for their choice of Bayesian analysis by referring to alternative methods used to drill down into large radiocarbon datasets. For example, Activity Event analysis is receiving growing attention in northwest Europe (e.g. Wicks & Mithen 2014, Waddington & Wicks 2017, Mithen & Wicks 2018). AE analysis is a means by which to extrapolate actual patterns in human histories indicated by the use of Summed Calibrated Probability Distributions (n.b the authors use the term 'Summed Probability Density Functions' – the distinction being simply a matter of pedantics). Taking this approach, calibration artefacts can be evaluated, along with biases introduced by well-dated sites yielding dates that are tightly aligned chronologically – these introducing false sharp peaks in SCPDs/SPDFs.

Furthermore, the authors may wish to explain the extent to which they screened their radiocarbon datasets for chronological hygiene i.e. how confident are they that the dataset used to map human activity can be reliably associated with the archaeological record? Deriving a radiocarbon date directly from an animal bone of an extinct fauna presents no problems here, whereas dating a charred hazelnut shell from an archaeological site demands a clear understanding of its stratigraphic relationship with the cultural remains it is being used to date.

My final comment is that when this paper lands, I anticipate it may well receive considerable media interest. Fascinating read, so thank you for requesting that I provide a review.

Signed: Dr Karen Wicks

Reviewer #3:

Remarks to the Author:

Introduction: The paper needs to lay out clearly what the hypotheses are and what results would

support Ho or H1, and why. Why would a correlation between human population density and megafaunal density support overkill? An argument could be raised that an inverse correlation might support it (areas without humans – megafauna unaffected, areas with humans, megafauna have been killed off).

Page 1/line 51: 10,000 is too rough an approximation for 11,700

2/32: suggest 'pre-date' rather than 'predate' to avoid confusion with predation

2/39-41: Regarding a 'continuing debate' about extinction dates, is *Coelodonta* a good example? It is true that a debate was aired in ref 23 (a 2013 paper), but my reading of it was that one party to the debate effectively quashed the criticisms of the other. Sometimes a debate can actually be settled, e.g. by showing that putatively very late radiocarbon dates were objectively unreliable (as in the woolly rhino case). (This is not to deny the logic that no FAD/LAD can be known perfectly). The apparent demonstration by Haile et al that mammoths survived longer in N America than any dated bone, from finding its aDNA in an indirectly dated horizon, might be a better example of genuine uncertainty.

4/14-22. This is my major problem with the paper. After giving a long and detailed explanation of the methods you dismiss, you then give only a few lines to the one you prefer. I could not fathom what was meant by "one probable event-count sequence from the universe of possible sequences defined by the joint density of all the relevant radiocarbon dates". Turning to the Methods for enlightenment, I continued to struggle. Here are some indications of the problem:

17/13-14: define 'event count sequence'

17/14: define 'hyperparameter'

17/15: define an 'event', as in 'individual event times'. I assume it means the death of an animal that has then been dated, but this should be clarified.

17/16-17: what does it mean to 'randomly sample a possible date'? How can you sample from one item? Or do you mean randomly take one date from several dates available for an event? But you have already excluded multiple dates for an event. Or are you talking about sampling one point from the 95% confidence interval (probability density function)?

17/20-21: 'number of events falling into each interval' - so you're turning the pdf of a calibrated date into a sort of histogram?

17/22: what is a 'member'? Is it one of the 'intervals'?

Even when Carleton et al. is published, readers of one paper should not have to read through another whole paper in order get a basic comprehension of the methods used.

Finally, I didn't find anywhere (apologies if I missed it) an explanation of WHY this method is better as a proxy for species population size than the summing of pdfs that you dismiss.

Two further significant points regarding methods:

17/9-11. The NGRIP record is used as a climate proxy in this study with no discussion. Considering that climate causality based on this proxy is the crux of the entire paper, please explain why air temperature above Greenland is a good proxy for climate across continental North America (and not just by saying it is 'commonly accepted' or similar).

4/34-36 Correcting for taphonomy by some assumed linear or curvilinear inverse relationship between preservation and time is a very dubious exercise in my opinion. It implausibly assumes a simple time-depth relationship between age and preservation. In geologically very recent sequences like the Late Pleistocene, factors like fluctuating climate and all its effects are far more important, e.g. an EARLIER

temperate episode with rivers flowing and depositing would preserve more bones than a LATER cold one where all the depositional environments were frozen up. I would not consider this an 'established' protocol and even if it is you should justify its use and, importantly, run the analyses without the correction. To what extent are the results sensitive to the parameters of the taphonomic model? Possibly, the authors have indeed done this, but the text on the matter is obscure: "Interestingly, the taphonomic proxy (β TA) appears to have had no effect in any model either" (6/13). Does that mean there was no difference in results with or without applying the taphonomic correction?

18/19-27: I also didn't understand the way in which the taphonomic correction was estimated. What is a 'tephra event count sequence'?

Some other comments on Methods:

16: The overall auditing protocol seems good.

16/27: does 'contiguous' exclude Alaska, and if so, explain why.

16/37-38: I assume from the Paisley cave example that by "where dated megafauna overlapped in their ages" you mean "where dated fossils of the same species from the same locality overlapped in their ages"? Please specify.

17/1: Does 'limited to 15-10 ka' mean limited to dates whose medians are in the 15-10 ka range, or any dates whose 95% range overlaps 15-10 ka?

17/4: 'Anomalous' dates were removed. I hope this doesn't mean you removed dates that just don't happen to fit some preconceived notion of when, for example, humans are 'expected' in a certain area? If it means one date is way different from others from the same level at a site, OK. Please define 'anomalous' - even if you took your cue from CARD, you need to justify it yourself.

17/7-8: "The vast majority of sites are evenly represented in respect to their radiocarbon record". I don't understand what this means.

Climate change, not human population growth, correlates with Late Quaternary megafauna declines in North America

Mathew Stewart, W. Christopher Carleton, Huw S. Groucutt

Response to reviewers

Reviewer #1

This is an excellent manuscript. It is a thoughtful and well-executed study that contributes to our understanding the timing and cause(s) of the extinction of North American Megafauna –a topic of critical importance to researchers focused on the Late Quaternary. In addition, this paper’s critique of use of summed probability density functions (SPDF) and their application of regression analysis with Radiocarbon-dated event count (REC) modeling means this paper has the potential to make significant methodological contributions to numerous fields of study. Its overall conclusion that the decline in North American megafauna is best explained by a changing climate, rather than an increase in human populations, is reasonable and generally fits other lines of evidence which are well- summarize in the introduction and discussion sections of this paper. Readers of Nature Communications will likely be highly interested in this paper. This said, there are a few relatively minor revisions to its data presentation and its evaluation of its key findings that should be made before final publication.

We thank the reviewer for their positive comments regarding our study.

The use of Radiocarbon Event Count Ensembles (RECEs) represents a useful new analytical tool. However, I feel the description of this analysis and how these data should interpreted needs more clarification, especially on page 4 (lines 14-19) and page 17 (as well as in the supplementary information). Including the 2020 JQS article in the supplemental material helped in understanding of this technique. But the average reader will benefit from more discussion of this procedure in the main paper. The authors should also more carefully explain the interpretations of Figures 5, 7, and 8 (and similar graphics in the supplementary data) where RECE counts are presented. I do not think it is entirely clear to the reader how to identify the “higher density regions (darker areas)”. In addition to a better description of this analysis, these figures might also be improved if a gradation of colors was used (such as “heat maps”) to depict the RECE rather than just using black and grays. The RECE lines are particularly hard to see in the less dense areas or when counts rise to 2 or 3.

We now discuss in greater detail the methods used for the analyses in the Introduction and Methods sections.

To help readers interpret the RECE figures, we have produced an additional “how to read a RECE model” figure (Fig. S9). This figure presents a zoomed in (25 year) portion of the human RECE model. In addition, we have added in a scale of “RECE Agreement” and, as an example, have highlighted a single year (11704-year BP) and labelled the degree of agreement (i.e., the amount of overlap between each of the RECEs). As shown, for the year 11704 BP, 171 RECE members produced a count of one (high agreement), 20 produced a count of two (moderate agreement), none produced counts of three and four (high agreement that the count was not

three or four in that year), and a single RECE produced a count of five (low agreement with the others). We have provided a detailed caption to help readers interpreted Fig. S9.

We have changed the scheme of the RECE count figures from greyscale to a colourmap, with dark colours denoting areas of low RECE agreement (i.e., regions on the figure where there is a low degree of agreement [overlap] between each of the RECEs) and bright colours denoting areas of high RECE agreement (i.e., regions on the figure where there is a high degree of agreement [overlap] between each of the RECEs). Also, this colour scheme increases the contrast between areas of low RECE agreement—such as where counts rise to 2 and 3—and the figure background, making the figure much easier to interpret. The Figure 5 caption now reads as:

“...RECEs are plotted in colourmap (magma) so that higher (brighter colours) and lower (darker colours) density regions can be visually distinguished. Clovis period delineated by the black box.”

In addition, while results of the regression analyses (e.g., figure 2-4 & 6) are pretty straightforward, I did wonder whether the reader should place any importance on the fact that the regression coefficient (β_{climate}) values for Sabertooth and Equus appear to be significantly higher than the other taxa? Based on Figure S9, Sabertooth and Equus appear to go extinct earlier than the other taxa. I do note that the brief discussion on page 14 (lines 34-39) seems to present a slightly different interpretation of Figure S9—which possibly relates to my comments above. In any case, if we accept the conclusion that climate change is the predominate variability influencing megafauna extinctions should there not be more consideration of the nature of the climate change at the time of these species disappear?

Again, we thank the reviewer for their suggestion regarding the colour scheme of the RECE figures. This has made the human and megafauna population density fluctuations much easier to read, and this has slightly changed our interpretation for two taxa (i.e., horse and saber-tooth). The final horse population decline appears to have occurred slightly earlier (i.e., during the terminal Bølling-Allerød) than previously suggested, whereas the final saber-tooth population decline may have occurred slightly later (i.e., closer to the Younger-Dryas boundary). We have amended the text as follows:

“The same can be said independently for mastodon, saber-tooth, and sloth populations, whereas final mammoth population declines appear to have occurred later in the YD, and final horse population decline may have occurred during the terminal B-A (Fig. S9).”

There are two immediately apparent explanations as to why the regression coefficients (β_{climate}) for horse and saber-tooth cat are higher than those of other taxa. The first relates to data quantity. Of the five taxa included in this study, horse and saber-tooth cat have the smallest sample sizes. It's possible that an increased sample size would decrease the observed slope, bringing the regression coefficient values more in line with the other taxa in this study. In our extended analysis, for which we conducted additional data cleaning and incorporated chronological uncertainty in our climate proxy, the regression coefficients are more consistent across taxa (Fig. 6).

The second explanation is that the analysis has accurately identified a stronger (positive) relationship between the population densities of these two taxa and climate change—i.e., populations of horse and saber-tooth cat may have been more sensitive to temperature changes (and associated ecological changes) than the other taxa in our study. However, we think that any comment on this would, at present, be highly speculative. Ecological changes during the latest Pleistocene were both temporally and spatially variable, and our understanding of these is hampered by the coarse resolution of, and sometimes disparate, palaeoecological proxies. This is further complicated by our limited understanding of the physiology (e.g., thermal tolerances) of these extinct taxa, often relying on our knowledge of close living relatives (if such taxa exist). And lastly, precisely when these species went extinct is not clear given the issues associated with LADs, as discussed in the Introduction.

Instead, we think the more important take-home message of our study is that no matter how the data are divided (all megafauna combined, by taxa, or by region) the results are consistent. Our data show that megafauna population numbers trended with climate change—i.e., generally speaking, as temperatures increased, megafauna populations levels increased, and *vice versa*. Also, our data are consistent in showing that the final megafauna declines occurred around the onset of the Younger Dryas, or in the case of horses perhaps during the terminal Bølling-Allerød following a lengthy period of temperature decline. And lastly, all megafauna extinctions occurred during or shortly after the close of the Younger Dryas. Given the spatial variability in climate and ecological changes during the Late Quaternary, we focus our discussion on the region-specific aspects of the study for which changes in ecology are more specifically known.

Understanding the more nuanced aspects of North American megafauna extinctions will require much more data. Still, given the largest database available to-date, our findings suggest that global decreases in temperature played a key (proximate) role in megafauna extinctions.

Overall, the discussion section of this paper is thorough and thoughtful. However, I believe the authors have data that could be used to explore some of the alternative hypotheses more deeply. For example, on page 13 (lines 35-51), there is a brief discussion of how the loss of keystone megaherbivores and interspecific competition among carnivores and humans may contribute to megafauna extinction in the absence of increasing human populations. This paper has data (for example figure S9) which seemingly could be leveraged to consider these alternatives in more detail. Mammoths and Mastodons are widely discussed to be keystone species, so what is the implication of them surviving longer than the other species considered in this paper? Likewise, if Sabertooth disappeared centuries before evidence for increase human populations does that have any implications for interspecific competition?

In response to this comment, we have extended our discussion on our data and its implications for the alternative hypotheses put forth in the paper. We agree with the reviewer regarding our data and the keystone megaherbivore hypothesis, but note that megaherbivores needn't go extinct to have a significant impact on ecology. Still, it is interesting that horse and, in particular, saber-tooth cat population densities significantly decline at a time when megaherbivore numbers were increasing. We have added a section on this:

“While this may have been the case for megafauna more broadly, our data indicate that at least some species of megafauna declines occurred prior to declines in keystone

megaherbivores. Specifically, final declines in horse and saber-tooth cat population densities significantly pre-dated those of mammoths and mastodons. In fact, these population declines occurred at a time of increasing mammoth and mastodon numbers, which is particularly interesting in the case of the saber-tooth cat, which is often considered to have been a specialised hunter of these very large animals (Ripple and Van Valkenburg, 2010)."

The final decline in saber-tooth cat populations is less straight-forward than for other taxa, in part due to the smaller sample size. Interestingly though, there does appear to be a decline in saber-tooth numbers coinciding with the emergence of Clovis-point wielding people in the Americas suggesting that competition with humans may have had an initial impact on saber-tooth populations. However, the final saber-tooth population decline appears to occur later during the early stages of the Younger Dryas. We have added the following section:

"Interestingly, there does appear to be a drop in saber-tooth cat population density coinciding with the emergence of Clovis-point wielding peoples in the Americas suggesting that inter-specific competition may have had an initial impact on saber-tooth cat populations; although, the sample size for saber-tooth cat is rather small, and the final population decline appears to occur later and closer to the Younger Dryas."

Also, the data from mammoth and mastodon suggest that if declines and extinctions were driven by habitat fragmentation, that this occurred not during the emergence of Clovis but much later during the Younger Dryas.

"If so, the mammoth and mastodon data suggest that this occurred not with the arrival of Clovis-point wielding people, but much later during the YD."

Finally, the authors choice to use Broughton and Weitzel dataset for North American 521 megafauna is reasonable and justifiable given its size and quality. Further, the authors use of a modified subset of just 432 specimens also is reasonable given their concern about duplication of samples. However, the supplementary data they provide appears to be simply a copy of the Broughton and Weitzel's original data. I would suggest adding a column in this file which identifies the samples dropped from consideration, or perhaps simply limit the table to just the specimens included in the analysis. It also would be simple for add that the authors to add 20 or so additional Mastodon dates, which have been recently published in Karpinski, E., Hackenberger, D., Zazula, G. et al American mastodon mitochondrial genomes suggest multiple dispersal events in response to Pleistocene climate oscillations. Nature Communications 11, 4048 (2020).

A column indicating which dates were dropped following our extended chronometric cleaning has been added as a supplementary data file.

We thank the reviewer for bringing our attention to the new dates presented in Karpinski et al. (2020). We queried their dataset for mastodon dates that fit within the temporal and geographic parameters of our study and excluded those already included in our analysis. The result was an additional seven dates. Using these dates, we ran two extended analyses. In one, we compared the updated mastodon sample to the human and climate proxies using the same analytical procedure described in our main paper. This involved all of the original mastodon dates plus the seven new ones. In the other analysis, we extracted the mastodon samples from the Great Lake's region (including the new ones from Karpinski et al. 2020) and compared those to the

human and climate proxies following the procedures described in our paper for the regional analyses. This one involved all of the Great Lakes region mastodon fossil dates we had already plus five of the new dates.

Our results were consistent with all our previous findings—i.e., there was no correlation between through-time changes in human and megafauna population densities, but there was significant positive correlation between megafauna population densities and our climate proxy (see Fig. 1 below).

Figure 1: Regression results for the mixed human and climate models for mastodon across the contiguous US (top) and Great Lakes region (bottom). Data includes the new dates from Karipinski et al. (2020). As can be seen, the results are consistent with our earlier findings. That is, there is a through-time significant, positive correlation between mastodon population density and our climate proxy, but no correlation between mastodon and human population densities

We have decided, however, that it is best to exclude these additional analyses from the revised manuscript. As currently presented, our paper works with an existing radiocarbon dataset, which we further cleaned for chronometric hygiene. Since the publication of this dataset in 2018, there is certainly a small number of new radiocarbon dates which could be added to the existing dataset, and more radiocarbon dates will become available as research on this topic progresses. On that note, we think a good idea would be to revisit this once enough new radiocarbon dates are available to warrant re-running the analyses. For now, the Karpinski et al. (2020) mastodon dates have been added to our dataset intended for future analysis, and we retain the original dataset to ensure that our study is comparable to other recent studies on the topic (i.e., Broughton and Weitzel, 2018).

Reviewer #2

Cutting straight to the chase, I have no hesitation in recommending that this paper is published in the esteemed Nature Comms journal. For the Editor and the Authors, to place my few comments in context, I have a background in using Bayesian statistics to study Late Glacial and Mesolithic radiocarbon datasets obtained from archaeological sites to reconstruct human population demographics in northern Britain. As such, some of the territory covered by this paper is unfamiliar study ground in terms of its geographical range and faunal histories. That said, I hope my few comments below will provoke some thought concerning alternative

approaches to analysing radiocarbon datasets. They are by no means posited to stymie the robust statistical analysis undertaken by the authors.

This is a stand-out paper providing an interpretation of radiocarbon data using Bayesian statistical methods to show that megafaunal extinctions in North America are chronologically aligned with climate change events occurring at the Late Pleistocene-Early Holocene transition. Its findings challenge alternative hypotheses attributed to human-related impacts. This paper will be of significance to a wide community of archaeological prehistorians, ecologists and palaeogeographers with interests in climate change, extinction events and human demographic studies ranging across the Americas, Eurasia and Antipodean continents. The authors have taken a novel approach not seen frequently (if at all) in similar studies, by using a Bayesian statistical analysis – that of so-called Radiocarbon Dated Event-Count (REC) Modelling.

We appreciate the supportive comments.

I recommend that the authors expand the rationale for their choice of Bayesian analysis by referring to alternative methods used to drill down into large radiocarbon datasets. For example, Activity Event analysis is receiving growing attention in northwest Europe (e.g. Wicks & Mithen 2014, Waddington & Wicks 2017, Mithen & Wicks 2018). AE analysis is a means by which to extrapolate actual patterns in human histories indicated by the use of Summed Calibrated Probability Distributions (n.b the authors use the term ‘Summed Probability Density Functions’ – the distinction being simply a matter of pedantics). Taking this approach, calibration artefacts can be evaluated, along with biases introduced by well-dated sites yielding dates that are tightly aligned chronologically – these introducing false sharp peaks in SCPDs/SPDFs.

In response to this comment, we have expanded our methods section. We now explain the model we used in more detail, which will clarify our choice for the reviewer (and other readers, of course). We have also cited the reviewer’s work and explained why we did not use the AE approach with respect to discerning distinct events in our supplementary analysis where we aggressively filtered out overlapping densities—in short, it wasn’t necessary, though we think the approach is a good one.

The relevant revised sections of text are as follows:

“Therefore, while SPDFs may be helpful tools for summarizing chronological information or discerning certain patterns in large radiocarbon date databases (e.g., Wicks and Mithen, 2014), they are not an unambiguous indication of event-count or, by extension, a suitable proxy for population levels in a point-wise way.”

“We reasoned that any major differences in findings between our analysis involving the whole dataset and the analysis involving the aggressively filtered one would have indicated that potentially important biases might have been present in the data and that a more nuanced approach to event identification would be necessary (e.g., Wicks and Mithen, 2014). The results, however, were consistent with our other findings.”

“With this in mind, we used a recently developed alternative—Radiocarbon-dated Event Count (REC) modelling³³—to evaluate the North American megafauna overkill and climate change hypotheses. The new approach is a Bayesian regression technique that accounts for

chronological uncertainty in time series of radiocarbon-dated event counts. It involves sampling alternate probable count sequences that are consistent with the uncertainties in the individual radiocarbon-date densities in a given database. A sample of alternate sequences---a Radiocarbon-dated Event Count Ensemble (RECE)—is first produced. Each sequence in the sample (RECE member) is then used as the response variable in a suitable regression model. The parameters estimated for these individual models are considered to be samples from a set of super-population parameter distributions that reflect the variability among the individual regression estimates. These individual model estimates vary because the alternate count sequences are all slightly different, reflecting the chronological uncertainty in the corresponding radiocarbon dates—a sequence of fossil counts might be {1,2,3} or {2,1,3} depending on the sample of probable dates for fossils in the dataset. Thus, the super-population parameters of the model reflect chronological uncertainty as well. In effect, the REC model considers alternate histories, given the uncertainty in radiocarbon dates, and it uses those alternatives to estimate a set of super-population parameters (e.g., regression coefficients) that are consistent with the set of alternate histories (see the Methods section for further details).”

To respond further to the reviewer’s comments here, though, the kind of analysis we needed to undertake could not be conducted with an SPD[F] or a histogram of binned median dates (however unique events have been identified). SPD[F]s conflate process variation and chronological uncertainty in such a way that these kinds of information are inseparable and, as a result, formal regression models are mis-specified. This leads to biases and uncontrolled error rates of both main types (false positives and negatives). Importantly, this would be the case even if uncalibrated summed densities were used, which would remove the “artifacts” created by calibration entirely. Essentially, the problem boils down to treating densities (one or many summed together) as if they represent through-time variation in event-count. No amount of smoothing or wiggle reducing eliminates the fundamental problem.

The date binning approach (part of the reviewer’s AE methodology), while maybe suitable at coarse scales/resolutions for identifying certain deviations from a specified null process, cannot be used for formal variable selection (comparing the potential quantitative impacts of one or more covariates on an observed time series). Additionally, depending on the (arbitrary) choice of bin-width, binned median dates would produce a highly biased impression of through-time patterns because the chronological uncertainty in individual event-times would be left out of the analysis. REC models can account for the uncertainty, “rolling it into” the regression parameter estimates (posterior densities).

Furthermore, the authors may wish to explain the extent to which they screened their radiocarbon datasets for chronological hygiene i.e. how confident are they that the dataset used to map human activity can be reliably associated with the archaeological record? Deriving a radiocarbon date directly from an animal bone of an extinct fauna presents no problems here, whereas dating a charred hazelnut shell from an archaeological site demands a clear understanding of its stratigraphic relationship with the cultural remains it is being used to date.

In response to this comment, we have included in more detail the data cleaning criteria used (see Methods section), and below we provide our rationale for maintaining the dataset as presented by Broughton and Weitzel (2018). We have also added an additional column to the supplementary dataset indicating which samples were removed following our additional cleaning.

The archaeological dataset was first screened by Broughton and Weitzel (2018) by removing: (1) duplicates; (2) dates marked as ‘anomalous’; (3) dates derived from non-anthropogenic contexts; and (4) dates derived from megafauna but lacking clear butchery/kill associations. This initial screening served two key purposes. Firstly, criteria 1 & 2 cleaned the dataset for quality—for example, by removing dates obtained through non-modern techniques. Secondly, criteria 3 & 4 ensured that the data being analysed could be confidently associated with the archaeological record by including only dates derived from archaeological contexts and megafauna with unambiguous evidence of anthropogenic processing. Using this dataset ensured that our study was comparable with other recent studies in the use of this proxy (e.g., Broughton and Weitzel, 2018).

In addition, we also looked at the distribution of sites containing a given number of radiocarbon samples to see whether the dataset might be biased by intensively dated sites (Fig. S10 in original submission). As shown in Fig S10, the majority of sites are evenly represented with respect to the number of radiocarbon dates, with most sites being represented by only one or two dates. Therefore, the dataset is not biased by intensively dated sites and instead could be considered to reflect counts of individual sites.

All this said, and as pointed out by the reviewer, understanding the precise relationship between dated material and cultural material at a particular site requires a comprehensive understanding of that site’s excavation, stratigraphy, and taphonomy. This is, however, an unrealistic goal in studies that comprise hundreds of sites (such as ours), and one must consider the trade-offs between using smaller, more aggressively vetted datasets and larger datasets more amenable to sophisticated statistical analysis. There are two other points to be made here. First, in analyses such as these, some assurance must be extended to those responsible for the data entries. These are people often directly involved in and/or are intimately familiar with the research. Data entries are often accompanied with detailed comments on the site stratigraphy, material dated, and dating methods. Dates ‘anomalous’ in age (e.g., dated material too young to be associated with the cultural material with which it was found) are often flagged and can be easily removed prior to analysis (as was done in our dataset). Secondly, dated material (e.g., a charred hazelnut shell) and cultural material which has incorrectly deemed to be associated is only likely to have impact the analysis if (a) the issue is prevalent and (b) it is non-random (i.e., would have to systematically bias the dataset). We consider neither of these two likely to be a significant issue.

My final comment is that when this paper lands, I anticipate it may well receive considerable media interest. Fascinating read, so thank you for requesting that I provide a review.
Signed: Dr Karen Wicks

Again, we thank the reviewer for their positive and constructive comments.

Reviewer #3

Introduction: The paper needs to lay out clearly what the hypotheses are and what results would support Ho or H1, and why. Why would a correlation between human population density and megafaunal density support overkill? An argument could be raised that an inverse correlation

might support it (areas without humans – megafauna unaffected, areas with humans, megafauna have been killed off).

Simple overkill models hypothesize that rapidly expanding human populations—expanding both in terms of population size and geographic area—drove megafauna declines and extinctions. As such, under the overkill model, human and megafauna populations are expected to be inversely correlated (i.e., significantly and negatively correlated). We have expanded on our expectations in the Introduction section:

“Put differently, if human overkill drove megafauna extinctions, we expect there to be a negative and statistically significant (non-zero) correlation between the human and megafauna population density proxies. Likewise, if rising temperatures drove megafauna extinctions, we expect a negative and statistically significant correlation between our megafauna population density and climate proxies, or, alternatively, if decreasing temperatures caused megafauna extinctions, a positive correlation between these two proxies.”

Our results show that there is no correlation (either positive *or* negative) between human and megafauna population densities, and, therefore, no support for overkill given the available data.

Page 1/line 51: 10,000 is too rough an approximation for 11,700

10,000 years BP has been changed to 11,700 years BP.

2/32: suggest ‘pre-date’ rather than ‘predate’ to avoid confusion with predation

Amended.

2/39-41: Regarding a ‘continuing debate’ about extinction dates, is *Coelodonta* a good example? It is true that a debate was aired in ref 23 (a 2013 paper), but my reading of it was that one party to the debate effectively quashed the criticisms of the other. Sometimes a debate can actually be settled, e.g. by showing that putatively very late radiocarbon dates were objectively unreliable (as in the woolly rhino case). (This is not to deny the logic that no FAD/LAD can be known perfectly). The apparent demonstration by Haile et al that mammoths survived longer in N America than any dated bone, from finding its aDNA in an indirectly dated horizon, might be a better example of genuine uncertainty.

After re-reading ref 23, we agree with the reviewer that the “debate” regarding the LAD of *Coelodonta* is more settled than originally indicated in our article. We thank the reviewer for pointing us to the paper by Haile and colleagues (2009) and agree that this represents a better example of the issues of using LADs based on dated fossil remains. We have changed this and the section now reads as follows:

*“There are, however, problems with FAD- and LAD-based studies. For instance, the LAD of *Smilodon fatalis* is, with near certainty, not derived from remains of the last living saber-tooth cat—a phenomenon known as the Signor-Lipps effect²². Even for extensively dated taxa, such as mammoth (*Mammuthus primigenius*), aDNA studies have shown the survival of taxa far beyond LAD’s based on dated fossil remains²³.”*

4/14-22. This is my major problem with the paper. After giving a long and detailed explanation of the methods you dismiss, you then give only a few lines to the one you prefer. I could not

fathom what was meant by "one probable event-count sequence from the universe of possible sequences defined by the joint density of all the relevant radiocarbon dates". Turning to the Methods for enlightenment, I continued to struggle. Here are some indications of the problem:

We have revised the overview of REC models in the Introduction and expanded the Methods section substantially to better explain the approach.

17/13-14: define 'event count sequence'

Definition of 'event count sequence' added:

"In the present case, "event" refers to the death of an animal, and a corresponding count-based time series (event count sequence) would ideally indicate through-time changes in the number of animal deaths over a given period."

17/14: define 'hyperparameter'

Definition of 'hyperparameter' added:

"...hyper-parameter distributions—where a "hyper-parameter" is a parameter of a distribution that characterizes the uncertainty in another model parameter (the mean of a distribution of other means, for example)."

17/15: define an 'event', as in 'individual event times'. I assume it means the death of an animal that has then been dated, but this should be clarified.

This is correct. See comment above.

17/16-17: what does it mean to 'randomly sample a possible date'? How can you sample from one item? Or do you mean randomly take one date from several dates available for an event? But you have already excluded multiple dates for an event. Or are you talking about sampling one point from the 95% confidence interval (probability density function)?

Radiocarbon dates have an associated uncertainty. As such, their precise date is unknown and they must be reported as a distribution of possible dates—for example, a fossil may date to between 150–100 years old (within some confidence interval). The REC models randomly sample a possible date—in this case a single year—from within the possible age range and in accordance with the probability that the given event (i.e., the death of the animal) occurred on the relevant date.

We have amended the text to better explain this better:

"The events in question, however, are dated with radiocarbon assays. These date estimates have uncertainties associated with them, which are represented by distributions of possible dates that can span many centuries. A single event (animal death) might have occurred at any time within the domain (timespan) of the distribution corresponding to the radiocarbon-date from the relevant fossil. Thus, the true time-series of fossil counts cannot be established because individual fossils cannot be assigned to any given date with absolute certainty. Multiple potential event count sequences are, therefore, always possible."

17/20-21: ‘number of events falling into each interval’ - so you're turning the pdf of a calibrated date into a sort of histogram?

No. Our phrasing was confusing. We have amended the relevant text and expanded the Methods section to better explain what we did.

17/22: what is a ‘member’? Is it one of the ‘intervals’?

This is now clarified in the text:

“Each individual sequence in the RECE (a single member) represents one of the probable event count sequences that might have occurred in the past.”

Even when Carleton et al. is published, readers of one paper should not have to read through another whole paper in order to get a basic comprehension of the methods used.

We have included in greater detail the methods used for the analysis. Readers should now be able to fully comprehend the methods without turning to Carleton (2020).

Finally, I didn’t find anywhere (apologies if I missed it) an explanation of WHY this method is better as a proxy for species population size than the summing of pdfs that you dismiss.

Our expanded methods section now explains the advantages of the REC method in greater detail.

Two further significant points regarding methods:

17/9-11. The NGRIP record is used as a climate proxy in this study with no discussion. Considering that climate causality based on this proxy is the crux of the entire paper, please explain why air temperature above Greenland is a good proxy for climate across continental North America (and not just by saying it is ‘commonly accepted’ or similar).

We used the NGRIP record so that our study was comparable with other recent studies on megafauna extinctions (e.g., Broughton and Weitzel, 2018). Furthermore, the NGRIP record is typically considered to be a suitable proxy for long-term trends in global temperature fluctuations. Indeed, it matches well with other highly-resolved proxies from the Americas and Europe. We have added a sentence to explain these points and the relevant section now reads as follows:

“The NGRIP record matches well with other highly-resolved temperature proxies from North America and Western Europe (e.g., Shackleton and Hall, 2000; Wagner et al., 2010; Affolter et al., 2019), and, as such, is considered a suitable proxy for broad-scale long-term trends in late Pleistocene global—or at least Northern Hemispheric—temperature fluctuations.”

4/34-36 Correcting for taphonomy by some assumed linear or curvilinear inverse relationship between preservation and time is a very dubious exercise in my opinion. It implausibly assumes a simple time-depth relationship between age and preservation. In geologically very recent sequences like the Late Pleistocene, factors like fluctuating climate and all its effects are far

more important, e.g. an EARLIER temperate episode with rivers flowing and depositing would preserve more bones than a LATER cold one where all the depositional environments were frozen up. I would not consider this an 'established' protocol and even if it is you should justify its use and, importantly, run the analyses without the correction. To what extent are the results sensitive to the parameters of the taphonomic model? Possibly, the authors have indeed done this, but the text on the matter is obscure: “Interestingly, the taphonomic proxy (β TA) appears to have had no effect in any model either” (6/13). Does that mean there was no difference in results with or without applying the taphonomic correction?

The reviewer's confusion here stems from our lack of clarity. We did not apply a correction as recommended by Surovell et al. Instead, we included the tephra proxy they describe (and used to devise their “correction”) as a covariate in all the models, thereby accounting for taphonomic processes in our analyses. We explain this in more detail and justify the use of the tephra record as follows:

“Crucially, this proxy would be subject to any time-dependent and climate-dependent process that affect taphonomy on a regional scale. By including it as a covariate in our regressions, we allowed for the possibility that these taphonomic processes account for through-time variation in fossil counts. If the relevant regression coefficient was determined to be non-zero, then the taphonomic proxy would explain some of the variation in fossil counts thereby reducing the variation leftover for other covariates to explain. Alternatively, if it was estimated to be zero, it would indicate that regional, long-term taphonomic processes cannot explain variation in fossil counts.”

18/19-27: I also didn't understand the way in which the taphonomic correction was estimated. What is a 'tephra count sequence'?

Like the fossil data, the tephra record is composed of dated samples that represent deposition events. The material dated refers to the timing of tephra deposition (from volcanic regional activity). The associated dates have uncertainties, just like the fossil dates, and so we could include them in the models in the same way. These data were used to create RECEs for the tephra record (taphonomic proxy) and those were included as covariates in the Bayesian regression model. Importantly, as we explain in the revised text above, these records would have been subject to the same time-dependent and/or climate dependent taphonomic pressures as the fossil samples (at regional scales). If the relevant regression coefficient(s) differed significantly from zero in a given regression intended to explain variation in megafauna fossil counts, it would indicate that taphonomy accounted for at least some of the variation in fossil counts. That wasn't the case, as it turns out. But, even had it been the case, the remaining regression coefficients would have referred to the marginal effects of the other covariate(s) (i.e., human population size proxy and/or climate proxy), which means that the effect of taphonomy could be separated from those other effects. Essentially, a one-unit change in fossil counts would have been explained by a change in the taphonomy proxy + change in human pop. proxy + change in climate proxy (for the model involving all those covariates at once, at any rate).

Some other comments on Methods:

16: The overall auditing protocol seems good.

We are pleased to hear that the reviewer agrees with our additional data cleaning protocol.

16/27: does ‘contiguous’ exclude Alaska, and if so, explain why.

No. We excluded Alaska from our analysis following Broughton and Weitzel (2018) so that the two studies were comparable.

16/37-38: I assume from the Paisley cave example that by "where dated megafauna overlapped in their ages" you mean "where dated fossils of the same species from the same locality overlapped in their ages"? Please specify.

Exactly. We have amended this following the reviewers’ suggestion:

“As a first step, we flagged instances where dated fossils of the same taxon from the same locality overlapped in their ages (i.e., radiocarbon years before present (RCYBP) ± error).”

17/1: Does ‘limited to 15-10 ka’ mean limited to dates whose medians are in the 15-10 ka range, or any dates whose 95% range overlaps 15-10 ka?

The dataset includes any date whose radiocarbon date densities overlaps the 15-10 ka time interval. However, it is important to note that dates with only the tails of the date density falling within this interval are unlikely to be sampled in each REC model—i.e., each date is sampled according to its probability within the 15-10 ka interval. This allows the inclusion of all dates that feasibly could occur in this interval, while removing arbitrary edge affects that would result from the exclusions of any date whose median value falls just outside this interval.

17/4: ‘Anomalous’ dates were removed. I hope this doesn't mean you removed dates that just don't happen to fit some preconceived notion of when, for example, humans are 'expected' in a certain area? If it means one date is way different from others from the same level at a site, OK. Please define 'anomalous' - even if you took your cue from CARD, you need to justify it yourself.

To clarify, we ourselves did not remove ‘anomalous’ dates from the dataset. The data cleaning—which involved the (1) removal of duplicates, (2) anomalous dates, (3) dates derived from non-anthropogenic contexts, and (4) dates derived from megafauna but lacking clear evidence for a kill/scavenging association—was conducted by Broughton and Weitzel (2018). Our analyses were done using their already cleaned dataset.

The ‘anomalous’ dates were marked as such by uploaders to the CARD database. Dates may have been marked anomalous for a multitude of reasons (e.g., poor preservation, the use of non-modern dating techniques). Given that these were flagged by the uploader—in many cases people involved in the research or intimately familiar with it—it seems reasonable, if not sensible, to exclude these dates from the analysis. We provide one such example here from the Fletcher Site in southern Alberta:

“Possibly due to the influence of cattle urine and excrement, the site has been extraordinarily difficult to date by radiocarbon. The bones submitted for dating were somewhat permineralized, with manganese enrichment subsequently being demonstrated by neutron activation analysis (Wilson, et al. 1991: 130). Furthermore, the method of collagen extraction in use at the time employed the insoluble concentrate for radiocarbon dating, and this method

was subsequently abandoned when it was realized that potential contaminants might remain in the samples at variable concentrations.” (CARD database)

17/7-8: “The vast majority of sites are evenly represented in respect to their radiocarbon record”. I don’t understand what this means.

In this instance we are clarifying that most sites have a similar number of radiocarbon dates (i.e., a single radiocarbon dates) associated with them (at least as far as our analyses are concerned). This was important to check as too many oversampled sites may have biased our analyses. But, as shown in Figure S10, this appears not to be the case.

We have amended the sentence and hopefully this now reads clearer:

“The vast majority of sites are evenly represented in respect to their radiocarbon record—i.e., most sites in our analysis are represented by a single radiocarbon date (Fig. S10).”

Reviewers' Comments:

Reviewer #1:

Remarks to the Author:

In the initial review of this manuscript, I thought it a thoughtful and well-executed study that contributes to our understanding the timing and cause(s) of the extinction of North American Megafauna –a topic of critical importance to researchers focused on the Late Quaternary. My comments and suggestions on that version primarily focused on encouraging the the authors to expand their description of their RECE analysis and be more explanatory in how the reader should interpret these results. I also suggested that the authors should provide greater discussion of the differential results for Sabertooth and Equus compare to Mammoths and Mastodons. Finally, I made minor suggestions about their presentation of the Broughton and Weitzel dataset for North American 521 megafauna species.

The revised manuscript is a substantial improvement over the original. I felt that authors' responses to my comments and their changes to the manuscript were thorough and sufficient. In its current form I support the publication of this manuscript without reservation. I also commend the authors on their care and thoughtfulness in making these revisions.

Reviewer #2:

Remarks to the Author:

The authors have provided well argued (tho' debatable) responses to my comments - I hope I eventually get to meet with them at conference, so we have the opportunity to challenge each other's choice of statistical analyses to interpret radiocarbon date data.

I look forward to seeing this paper as it now reads, online and in print in due course.

Reviewer #3:

Remarks to the Author:

The authors have fully revised the ms in the light of all reviewers' comments. Indeed, their response document is one of the best I have seen in terms of explaining the issues and the revisions they have made. This paper will undoubtedly, as other reviewers noted, form an important milestone in research on Late Quaternary extinctions.

I have three requests for very minor but I think necessary final corrections before publication:

1. p. 2, Line 49. Please change 'shown' to 'suggested' or similar. The authors have misunderstood my request, that I see is my fault for lack of clarity. I was not suggesting that the DNA evidence of Haile et al conclusively shows that the mammoth LAD based on fossils was wrong; far from it. Subsequent authors have been sceptical about it, mainly because of the lack of direct dating, i.e. it assumes a link between DNA in a given horizon (mode of deposition unknown and potentially mobile [urine?]), and contextual dates for that horizon. It should not be posited as right or wrong; I was merely suggesting it as an example where evidence is *equivocal* (bone dates suggest one LAD, the DNA another).
2. Fig. 5 (and by extension Fig. 6) - the caption must indicate what 1-5 on the y-axis stands for.
3. P. 18, line 45: 'absent' is out of place here. Maybe 'without'?

Climate change, not human population growth, correlates with Late Quaternary megafauna declines in North America

Mathew Stewart, W. Christopher Carleton, Huw S. Groucutt

Response to reviewers

Reviewer #1

In the initial review of this manuscript, I thought it a thoughtful and well-executed study that contributes to our understanding the timing and cause(s) of the extinction of North American Megafauna –a topic of critical importance to researchers focused on the Late Quaternary. My comments and suggestions on that version primarily focused on encouraging the authors to expand their description of their RECE analysis and be more explanatory in how the reader should interpret these results. I also suggested that the authors should provide greater discussion of the differential results for Sabertooth and Equus compare to Mammoths and Mastodons. Finally, I made minor suggestions about their presentation of the Broughton and Weitzel dataset for North American 521 megafauna species.

The revised manuscript is a substantial improvement over the original. I felt that authors' responses to my comments and their changes to the manuscript were thorough and sufficient. In its current form I support the publication of this manuscript without reservation. I also commend the authors on their care and thoughtfulness in making these revisions.

We thank the reviewer for their original comments, which helped to greatly improve the manuscript, and their recommendation here for publication.

Reviewer #2

The authors have provided well argued (tho' debatable) responses to my comments - I hope I eventually get to meet with them at conference, so we have the opportunity to challenge each other's choice of statistical analyses to interpret radiocarbon date data.

We hope so too.

I look forward to seeing this paper as it now reads, online and in print in due course.

We thank the reviewer for their original comments, which helped to greatly improve the manuscript, and their recommendation here for publication.

Reviewer #3

The authors have fully revised the ms in the light of all reviewers' comments. Indeed, their response document is one of the best I have seen in terms of explaining the issues and the revisions they have made. This paper will undoubtedly, as other reviewers noted, form an important milestone in research on Late Quaternary extinctions.

We thank the reviewer for their original comments, which helped to greatly improve the manuscript, and their recommendation here for publication.

I have three requests for very minor but I think necessary final corrections before publication:

1. p. 2, Line 49. Please change 'shown' to 'suggested' or similar. The authors have misunderstood my request, that I see is my fault for lack of clarity. I was not suggesting that the DNA evidence of Haile et al conclusively shows that the mammoth LAD based on fossils was wrong; far from it. Subsequent authors have been sceptical about it, mainly because of the lack of direct dating, i.e. it assumes a link between DNA in a given horizon (mode of deposition unknown and potentially mobile [urine?]), and contextual dates for that horizon. It should not be posited as right or wrong; I was merely suggesting it as an example where evidence is *equivocal* (bone dates suggest one LAD, the DNA another).

Amended.

“Even for extensively dated taxa, such as mammoth (*Mammuthus primigenius*), sedimentary ancient DNA studies have suggested that some taxa survived far beyond their LAD’s based on dated fossil remains.”

2. Fig. 5 (and by extension Fig. 6) - the caption must indicate what 1-5 on the y-axis stands for.

We have added a description of the y-axis to the figure 5 caption:

“The y-axis represents the count—a count of two, for example, would result from two dated events occurring in the same year in a RECE member.”

3. P. 18, line 45: 'absent' is out of place here. Maybe 'without'?

Amended